# Comparison of Reanalysis and Observational Precipitation Datasets Including ERA5 and WFDE5

**Birgit Hassler *** and **Axel Lauer ***

Deutsches Zentrum für Luft- und Raumfahrt (DLR), Institut für Physik der Atmosphäre, Münchener Straße 20, 82234 Weßling, Germany
* Correspondence: birgit.hassler@dlr.de (B.H.); axel.lauer@dlr.de (A.L.);
  Tel.: +49-8153-28-2342 (B.H.); +49-8153-28-4247 (A.L.)

**Abstract:** Precipitation is a key component of the hydrological cycle and one of the most important variables in weather and climate studies. Accurate and reliable precipitation data are crucial for determining climate trends and variability. In this study, eleven different precipitation datasets are compared, six reanalysis and five observational datasets, including the reanalysis datasets ERA5 and WFDE5 from the ECMWF family, to quantify the differences between the widely used precipitation datasets and to identify their particular strengths and shortcomings. The comparisons are focused on the common time period 1983 through 2016 and on monthly, seasonal, and inter-annual times scales in regions representing different precipitation regimes, i.e., the Tropics, the Pacific Inter Tropical Convergence Zone (ITCZ), Central Europe, and the South Asian Monsoon region. For the analysis, satellite-gauge precipitation data from the Global Precipitation Climatology Project (GPCP-SG) are used as a reference. The comparison shows that ERA5 and ERA5-Land are a clear improvement over ERA-Interim and show in most cases smaller biases than the other reanalysis datasets (e.g., around 13% high bias in the Tropics compared to 17% for MERRA-2 and 36% for JRA-55). ERA5 agrees well with observations for Central Europe and the South Asian Monsoon region but underestimates very low precipitation rates in the Tropics. In particular, the tropical ocean remains challenging for reanalyses with three out of four products overestimating precipitation rates over the Atlantic and Indian Ocean.

**Keywords:** precipitation; ERA5; WFDE5; ESMValTool

## 1. Introduction

In meteorology, precipitation is usually defined as rain, snow, sleet, or hail falling towards the surface from a cloud. As a key component of the hydrological cycle, precipitation is of very high socio-economic importance and is relevant to many aspects of life [1]. Precipitation is therefore also one of the most important variables in weather and climate studies [2]. Accurate and reliable precipitation data are crucial for determining climate trends and variability [3] including impact relevant indices such as floods and droughts, and are important for the management of water resources, in the agriculture, forestry, and energy sectors, and also for weather, climate, and hydrological forecasting [4]. In particular, gridded precipitation datasets play an important role in evaluating and assessing the performance of global Earth system models (ESMs) (e.g., [5–7]) used for projections of future climate change [8]. Other examples of scientific applications of global precipitation datasets include the analysis of heavy rainfall over the Asian–Australian monsoon region [9], drought monitoring in China [10], or the diagnosis of rainfall over South America during strong El Niño-Southern Oscillation (ENSO) events [11].

Rain gauges are normally used to measure precipitation as point data directly at the Earth's surface [12]. It is, however, almost impossible to create a gap-free, long-term precipitation dataset from rain gauge data alone since the measurements are almost entirely available over land only, and there are regions where the density of measurement stations

is low or non-existent even today. In addition, local topographic features are known to influence rain gauge measurements, which can make calculations of precipitation values that are representative for large areas quite challenging. In contrast, satellite measurements provide more comprehensive spatial coverage and are not limited to land regions. A drawback, however, is their limited temporal coverage. Most satellite measurements of precipitation have only been available since the 1990s. Such datasets based on satellite measurements include, for instance, the TRMM-L3 dataset [13] or the PERSIANN-CDR dataset [14]. One of the most widely used measurement datasets in the climate science community today, however, are a combination of rain gauge and satellite measurements such as the GPCP-SG dataset [15].

Particularly in regions of very sparse instrumental coverage, reanalysis products are sometimes used as an alternative to observational datasets. For the production of modern reanalyses such as the European Centre for Medium-Range Weather Forecasts (ECMWF) reanalysis ERA5 [16], satellite, precipitation radar, and gauge measurements are assimilated. An advantage of reanalysis datasets is their globally complete and relatively long temporal coverage. A clear limitation is, however, that biases can be introduced by non-perfect models including, for instance, unresolved processes relevant to clouds and precipitation formation or uncertainties in used parameterizations and initial conditions. This is particularly relevant in regions with sparse observations such as over the oceans or in high-latitude regions where there is little effective constraint of the reanalysis solution and the fields are largely driven by the model physics and parameterizations (e.g., [17]).

Given the differences in measurement techniques between ground-based and satellite measurements, and the differences in creation between measurement datasets and reanalyses, it is maybe not surprising that different precipitation datasets show differences in both the magnitude and variability of precipitation estimates. Typically, reanalysis datasets show a larger degree of variability than the other types of datasets such as satellite- or rain gauge-based datasets with the degree of variability varying by region [2]. Large differences in annual and seasonal estimates between reanalyses and observations are found over the tropical oceans, in complex mountain areas, in northern Africa, and in some high-latitude regions [2]. It is therefore important to compare datasets from different sources and different providers. A number of such comparisons exist already, but they are typically either focused mostly on observational data (e.g., [2]), include only selected reanalysis and observational datasets (e.g., [18–20]), concentrate on specific continental regions such as, for example, the continental U.S. [21], North-Western Himalaya [22], or a river basin in China [23], or focus on selected ocean basins (e.g., the Southern Ocean [24]).

The analysis presented here includes a comparison of eleven different datasets consisting of six reanalysis and five observational datasets. Three of the datasets (ERA5-Land, GPCC and WFDE5), are only available over land. All analyses presented here are focused on monthly mean precipitation rate values, although for some datasets higher temporally resolved data are available. This study focuses on the Tropics, in their full coverage and also separated into land- and ocean-only regions, and two additional regions within the Tropics that are known for special precipitation regimes, including regions with known high biases. To be able to put the analysis of the Tropics in a general context, some global analyses and analyses focused on Central Europe are also presented. The aim of this study is to quantify the differences between the widely used precipitation datasets generated from different data sources and to identify their particular strengths and shortcomings including the recent reanalysis product ERA5 and the bias-corrected product WFDE5. The analysis focuses on monthly, seasonal, and inter-annual times scales.

The analyzed datasets are described in Section 2. Section 3 presents the methods and tools used. A comparison of the precipitation datasets including maps of precipitation climatologies, histograms, time series of area-averaged mean values and anomalies, and area mean annual cycle evaluations is presented in Section 4. In Section 5 the findings of the comparisons are discussed and summarized.

## 2. Data

The following sections provide an overview of the eleven datasets used in this analysis. The particular observational datasets were chosen since they are widely used in climate sciences (GPCP-SG, PERSIANN-CDR, GPCC [2,25–28]), or provide an independent precipitation estimate (TRMM-L3 [13] and E-OBS [29]). The reanalyses ERA5, ERA5-Land and WFDE5 were selected since they are quite recent reanalysis products and have not been extensively discussed in previous comparisons. ERA-Interim was added to the comparison since it is the predecessor of ERA5 and has been used extensively by the climate community (e.g., [30–33]). The datasets JRA-55 and MERRA-2 were included in the comparison as they are well established reanalysis products and they are produced by data providers other than ECMWF who produce the ERA5 and ERA-Interim products.

The main characteristics of the datasets are summarized in Table 1, including information about temporal coverage, spatial resolution, version number, and the main dataset reference. Datasets that have been used in [34] for the analysis and evaluation of climate models contributing to the Coupled Model Intercomparison Project Phase 6 (CMIP6 [35]) are marked with an asterisk.

**Table 1.** Summary of all analyzed datasets of total precipitation. Datasets that have been used in [34] for the analysis and evaluation of CMIP6 climate models are marked with an asterisk. All datasets have been used with a monthly time resolution.

| Dataset | Institution | Type | Time Range | Version | Observation Input (Relevant to Precipitation) | Resolution and Coverage | Main Reference |
|---|---|---|---|---|---|---|---|
| E-OBS | ECMWF | station data | January 1950–December 2019 | v21.0e-0.1 | station network of the European Climate Assessment & Dataset (ECA&D) | $0.1° \times 0.1°$ (Europe) | [36] |
| ERA5 * | ECMWF | reanalysis | January 1979–present | last access: 20 July 2020 | measurements from AMSR-2, AMSRE, GMI, SSM/I, SSMIS and TMI for cloud liquid water | $0.25° \times 0.25°$ (global) | [16] |
| ERA5-Land | ECMWF | reanalysis | January 1981–present | last access: 20 July 2020 | regridded ERA5 data, same observational sources as ERA5 | $0.1° \times 0.1°$ (global, land-only) | [37] |
| ERA-Interim | ECMWF | reanalysis | January 1979–December 2018 | last access: 5 September 2019 | none | $0.75° \times 0.75°$ (global) | [38] |
| GPCC * | DWD | station data | January 1891–December 2016 | V2018_025 | ~80,000 precipitation gauge stations world-wide with record durations of 10 years or longer | $0.25° \times 0.25°$ (global, land-only) | [39] |
| GPCP-SG * | GSFC/NASA | merged satellite + station data | January 1979–October 2017 | v2.3 (obs4MIPs) | microwave, infrared, and sounder data observed by the international constellation of precipitation-related satellites, and precipitation gauge analyses | $2.5° \times 2.5°$ (global) | [15] |

**Table 1.** *Cont.*

| Dataset | Institution | Type | Time Range | Version | Observation Input (Relevant to Precipitation) | Resolution and Coverage | Main Reference |
|---|---|---|---|---|---|---|---|
| JRA-55 | JMA | reanalysis | January 1958– December 2019 | obs4MIPs | primarily consist of observations used in ERA-40; from 1979: surface observations from fixed land stations (SYNOP) and upper-level observations used by NCEP/NCAR reanalysis | 1.25° × 1.25° (global) | [40] |
| MERRA-2 | NASA GMAO | reanalysis | January 1980– December 2020 | V5.12.4 | Measurements from SSM/I and TMI rain rate | Approx. 0.5° × 0.625° (global) | [41] |
| PERSIANN-CDR | NOAA CDR | processed satellite + station data | January 1983– December 2018 | v01r01 | ISCCP B1 IR data, GPCP v2.2 (merged to GridSat-B1) | 0.25° × 0.25° (approx. 60° S–60° N) | [14] |
| TRMM-L3 | NASA, JAXA | satellite | January 1998– December 2013 | 3B43 (obs4MIPs) | TMMR (PR, TMI, VIRS, CRES, LIS) | 0.25° × 0.25° (approx. 50° S–50° N) | [13,42] |
| WFDE5 | ECMWF | reanalysis | January 1979– December 2016 | v1.1-CRU+GPCC | ERA5 data, bias corrected based on the data from CRU TS 4.0 and GPCCv2020 | 0.5° × 0.5° (global, land-only) | [43] |

*2.1. E-OBS*

E-OBS is a daily gridded dataset with a high spatial resolution that covers the European region over land and is based on station data collated by the European Climate Assessment and Dataset (ECA&D) project. Currently, about 70 years of data from 1950 to present are available and the dataset is updated frequently. All station data are sourced directly from the European National Meteorological and Hydrological Services or other data-holding institutions. For a considerable number of countries, the whole national network of stations is used [44]. Most station time series are quality controlled by the respective agencies, but are also subject to further quality control following incorporation into ECA&D [36]. These data are then blended with time series from neighboring stations to form more temporally complete series [45]. The E-OBS dataset was used for research purposes including the validation and calibration of model results [29,46] and monitoring the climate across Europe [45]. Since version v18.0e, the dataset provides an improved estimation of the interpolation uncertainty obtained from of a 100-member ensemble of realizations of each daily field [36].

*2.2. ERA5*

The fifth generation of the ECMWF reanalysis (ERA5) replaces the highly successful ERA-Interim reanalysis [38]. ERA5 is based on four-dimensional variational (4D-Var) data assimilation using Cycle 41r2 of the Integrated Forecasting System (IFS), which is coupled to a soil model and an ocean wave model [47]. Here, ERA5 data were used that are served on the Copernicus Climate Change Service Climate Data Store (CDS) [47]. Atmospheric data are available on 137 hybrid vertical levels and are provided on the CDS interpolated to 37 pressure levels ranging from 1000 hPa (near surface) to 1 hPa (about 80 km) [48]. The monthly mean precipitation was calculated from the daily values that were accumulated from the sub-daily datasets [48]. More information about the ERA5 dataset can be found in [16].

### 2.3. ERA5-Land

ERA5-Land is an enhanced reprocessing of the land component using a higher resolution model version and taking ERA5 as an input, allowing a horizontal resolution of 9 km at hourly time steps [49]. Here, ERA5-Land data were used that are served on the CDS and are available on a regular $0.1° \times 0.1°$ grid, covering only land surfaces. Missing values are marked as such, so that zeroes can be interpreted as no precipitation [37]. The precipitation is interpolated from the same variable of the ERA5 product and used as a forcing field for the enhanced replays [49]. The monthly mean values from ERA5-Land provide averaged values of accumulated precipitation for each time step [50].

### 2.4. ERA-Interim

The ECMWF reanalysis ERA-Interim [38] is a global atmospheric dataset covering the time period from January 1979 through to August 2019. The dataset is commonly seen as the predecessor of the ERA5 reanalysis. The assimilation system used for ERA-Interim is based on Cycle 31r2 of the ECMWF IFS 2006 release, which includes a 4D-Var analysis. The dataset provides a temporal resolution of either 3 h (forecast) or 6 h (analysis) depending on the variable and is provided at a spatial resolution of $0.75° \times 0.75°$ (approximately 79 km) and on 60 vertical levels from the surface up to 0.1 hPa. More details are available at https://www.ecmwf.int/en/forecasts/datasets/reanalysis-datasets/era-interim (accessed on 1 September 2021) [51].

### 2.5. GPCC

The centennial Global Precipitation Climatology Centre (GPCC) is operated by Deutscher Wetterdienst (DWD). Its Full Data Monthly Product of monthly global land-surface precipitation is based on about 80,000 stations world-wide that provide data records of 10 years or longer [39]. It covers the time period from January 1891 through to December 2016. The data coverage per month varies from about 6000 (before 1900) to more than 50,000 stations. The Full Data Monthly Product is updated at irregular time intervals following significant improvements in the underlying data base. Monthly data are available on regular grids with different spatial resolutions ($0.25° \times 0.25°$, $0.5° \times 0.5°$, $1.0° \times 1.0°$, and $2.5° \times 2.5°$). Precipitation anomalies at the stations are interpolated and then superimposed on the GPCC Climatology V2018 in the corresponding resolution [39] instead of interpolating the absolute precipitation totals [52]. In this analysis, the dataset with a resolution of $0.25° \times 0.25°$ was used.

### 2.6. GPCP-SG

The Global Precipitation Climatology Project (GPCP) was established by the World Climate Research Program (WCRP) as part of the Global Energy and Water Cycle Exchanges (GEWEX). As a community-based analysis under the auspices of WCRP, GPCP was developed by an international consortium of researchers and operational scientists who provide datasets, products, and techniques [15]. The dataset covers the satellite era from 1979 to present on a global $2.5° \times 2.5°$ grid and is produced by merging a variety of data sources, including passive microwave-based rainfall retrievals from satellites (SSMI, SSMIS), infrared rainfall estimates from geostationary (GOES, Meteosat, GMS, MTSat) and polar-orbiting satellites (TOVS, AIRS), and surface rain gauges [15,53]. The most recent version 2.3 of GPCP-SG provides an improved homogeneity, especially in the time period since 2002 [15].

### 2.7. JRA-55

The Japan Meteorological Agency (JMA) provides a reanalysis dataset called JRA-55 (known as the Japanese 55-year Reanalysis) [40]. JRA-55 covers the time period from the beginning of global observations by regular radiosondes launches in 1958 to the present [54]. It is produced using the TL319 version (as of December 2009) of JMA's operational data assimilation system. The reanalysis is based on different observations in addition to the

45-year reanalysis data (ERA-40) provided by ECMWF. JRA-55 was the first comprehensive atmospheric reanalysis that applied 4D-Var analysis to this period [55]. Reduced model biases, improved dynamical consistency of analysis fields, and an extension the time period back to 1958 make JRA-55 suitable for studying multi-decadal climate variability and climate change [55].

### 2.8. MERRA-2

The Modern-Era Retrospective analysis for Research and Applications, Version 2 (MERRA-2) is produced by the National Aeronautics and Space Administration's (NASA) Global Modeling and Assimilation Office (GMAO) and provides data from 1980 to the present, using an updated version of the Goddard Earth Observing System Data Assimilation System Version 5 (GEOS-5 [56]) atmospheric general circulation model (AGCM) with a 4D-Var data assimilation scheme. MERRA-2 is the first long-term global reanalysis to assimilate space-based observations of aerosols and represents their interactions with other physical processes in the climate system [41]. The dataset contains hourly fields at a horizontal resolution of $0.625° \times 0.5°$ and 72 sigma vertical levels up to 0.01 hPa interpolated to 42 vertical pressure levels from 1000 hPa to 0.1 hPa.

### 2.9. PERSIANN-CDR

The Precipitation Estimation from Remotely Sensed Information using Artificial Neural Networks-Climate Data Record (PERSIANN-CDR) is a retrospective precipitation dataset based on multi satellite data, developed by the Center for Hydrometeorology and Remote Sensing (CHRS) and designed to be used for climate and hydrological studies. The dataset provides daily rainfall estimates on a $0.25° \times 0.25°$ grid between 60° S and 60° N from 1983 to the near present and is available through the CHRS Data Portal [57]. PERSIANN-CDR combines infrequent, but high-quality passive microwave (PMW) observations from lower Earth orbits with infrared (IR) observations from Geostationary Earth orbiting (GEO) satellites at high sampling rates [14]. Generally, algorithms are limited to the availability of the required input data. For the PERSIANN-CDR, neural networks are used to compensate the missing PMW data for the pre-1997 period to provide a high spatial resolution over four decades [58,59]. The primary input for the PERSIANN model is the Gridded Satellite IR (GridSat-B1) data from the International Satellite Cloud Climatology Project (ISCCP [60]). Monthly GPCP v2.2 data, which also contain surface rain gauge data [53], are used to calibrate the model at a $2.5° \times 2.5°$ resolution. PMW observations are used to update algorithm parameters. The bias-corrected PERSIANN precipitation estimates maintain a monthly total consistent with the monthly GPCP product [14]. The calibration of PERSIANN-CDR to data from GPCP introduces a dependency on the GPCP data although the datasets are not based on the same sources.

### 2.10. TRMM-L3

The Tropical Rainfall Measuring Mission (TRMM) is a joint mission between NASA and the Japan Aerospace Exploration Agency (JAXA) providing rainfall data for weather and climate research. The TRMM satellite launched in 1997 was equipped with a three-sensor rainfall suite, which includes a Precipitation Radar (PR), the TRMM Microwave Imager (TMI), and a Visible and InfraRed Scanner (VIRS) [13]. The TRMM satellite collected rainfall and lightning data for 17 years in the latitude belt 50° N to 50° S until the mission ended in April 2015. Postprocessing of the data, however, continued. The algorithm 3B43 was developed to produce the single, best-estimate precipitation rate and root mean square (RMS) precipitation error estimate field. For this, 3-hourly merged high-quality/IR estimates (which are also available as dataset TRMM-RT) are combined with ground-based radar data and a monthly accumulated rain gauge analysis (from GPCC) to apply a large-scale bias adjustment to the multi-satellite estimates over land. The combination of satellite and surface rain gauge data is similar to the GPCP-SG dataset [61].

*2.11. WFDE5*

The WATCH Forcing Data methodology applied to ERA5 (WFDE5) is provided at a horizontal resolution of $0.5° \times 0.5°$ and at a temporal resolution of 1 h covering the time period 1979–2016. It replaces the predecessor WFDEI [43]. WFDE5 has a higher spatial variability than its predecessor since it is generated by the aggregation of the higher-resolution ERA5 data [43]. The dataset was derived by applying the sequential elevation and monthly bias correction methods described in [62,63]. The bias correction for rainfall and snowfall flux is either based on the Climatic Research Unit dataset (CRU TS4.03 [64]), or based upon a combination of the dataset from CRU and that from GPCC v2018 [65]. In this study, the used data were bias corrected with both CRU and GPCC data. For simplicity, the WFDE5_CRU+GPCC dataset is from hereon called WFDE5. The total precipitation variable of WFDE5 was calculated as the sum of the rainfall and snowfall flux.

## 3. Tools and Methods

*3.1. Earth System Model Evaluation Tool*

All analyses in this study were made with the open-source community diagnostics and performance metrics tool for the evaluation of the Earth system models "Earth System Model Evaluation Tool" (ESMValTool [66–69]). For use with the ESMValTool, the variables and metadata of the precipitation datasets were reformatted following the CMOR (Climate Model Output Rewriter; https://pcmdi.github.io/cmor-site/media/pdf/cmor_users_guide.pdf (accessed on 18 June 2020) [70]) tables and definitions (e.g., https://github.com/PCMDI/cmip6-cmor-tables/tree/master/Tables (accessed on 7 November 2019) [71] for CMIP6). For this, the ESMValTool contains many scripts for reformatting observational datasets according to the CMOR standard, including scripts for ERA-Interim, WFDE5, JRA-55, MERRA-2, GPCC, E-OBS and PERSIANN-CDR. The scripts are publicly available on GitHub and are provided with the ESMValTool source code (https://github.com/ESMValGroup/ESMValTool (accessed on 1 September 2021)). The datasets GPCP-SG and TRMM-L3 are available from obs4MIPs [72–74] and can be used directly with the ESMValTool. "On-the-fly" reformatting for ERA5 and ERA5-Land were implemented in version 2 of the ESMValTool that allows the processing and analysis of the raw ERA5/ERA5-Land data without having to preprocess the data before running the ESMValTool.

All figures shown in this paper (apart from Figure 1) can be reproduced with the ESMValTool "recipe" recipe_mpqb_precip.yml, a configuration file defining input data, preprocessing steps, and diagnostics to be applied. The mean, correlation, and root-mean-square deviation (RSMD) values were calculated with a variation of the "recipe" recipe_lauer13jclim.yml.

*3.2. Geographical Regions*

Precipitation measurements have substantial uncertainties, especially if the precipitation amount is very small such as in the "drizzle" range. Meaningful comparisons with models or reanalyses are therefore mainly focused on large-scale patterns and climatologies based on monthly mean values rather than small-scale features on short time scales. The following comparisons include for this reason global maps of climatologies (multi-year annual means) and biases. Certain precipitation regimes are typically found in specific regions of the globe (see Figure 1). For these regions, histograms, mean time series, anomaly time series, and annual cycles are analyzed:

- Tropics: spans the latitude belt from 30° S to 30° N over all longitudes. In this region convection plays a dominant role and high temperatures allow for a high concentration of water vapor in the atmosphere. This region is typically associated with high precipitation values;
- Pacific Inter Tropical Convergence Zone (ITCZ): spans the region from 0° N to 12° N and 136° E to 85° W in the Pacific Ocean. In this region deep convection occurs frequently connected with large amounts of precipitation. Convection in the ITCZ is an important driver of the global circulation (Hadley cell);

- South Asian (SA) Monsoon: spans the region from 5° N to 30° N and 65° E to 95° E. In this region precipitation shows a distinct annual cycle with the frequent occurrence of heavy precipitation in summer (monsoon);
- Central Europe: spans the region from 42° N to 53° N and 0° E to 20° E. This is a region where many ground-based observations are available which are used for assimilation in reanalysis products. In the winter half-year precipitation is dominated by synoptic scale extratropical cyclones whereas in the summer half-year convective processes are dominant.

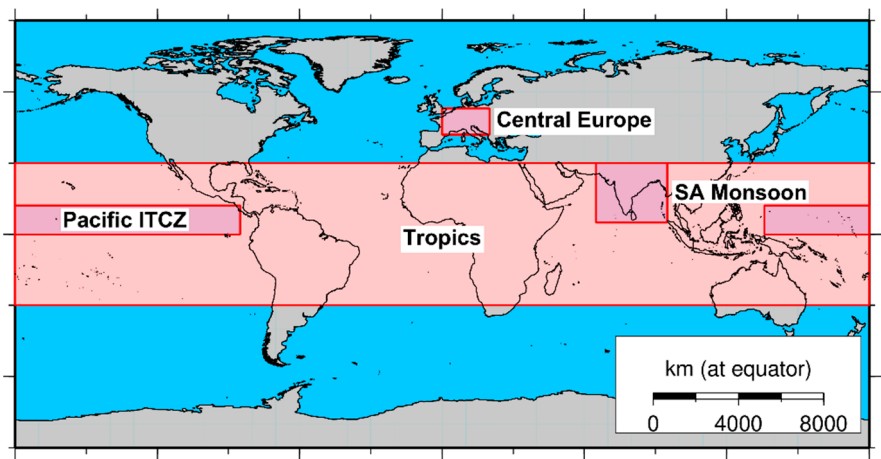

**Figure 1.** Definition of the geographical regions for which precipitation analyses were performed.

### 3.3. Regridding and Masking

When calculating the difference between datasets or correlations of a dataset with the reference dataset, all data were linearly regridded to the grid used by the GPCP-SG dataset (2.5° × 2.5°, see Table 1) using the regridding functions of the ESMValTool preprocessor [68]. Additionally, the application of land/sea masks, anomaly calculations, and area selection were also performed with the respective ESMValTool preprocessor functions. Grid cells were masked as missing if less than 80% of data from any of the datasets were available. Due to the missing values present in the ERA5-Land, GPCC and WFDE5 datasets over the ocean, and for the PERSIANN-CDR dataset at latitudes higher than 60°, three different "mean" and "correlation" values with respect to GPCP-SG were calculated in addition to the selected regions mentioned above: global with both land and ocean grid cells (ERA5, ERA-Interim, JRA-55, GPCP-SG, MERRA-2), global land-only (ERA5, ERA5-Land, WFDE5, ERA-Interim, JRA-55, MERRA-2, GPCP-SG, GPCC), and 60° S to 60° N with both land and ocean grid cells (ERA5, ERA-Interim, JRA-55, MERRA-2, GPCP-SG, PERSIANN-CDR).

## 4. Results

The analyses described in the next sections were mostly performed for the four regions described in Section 3.2, with the Tropics being analyzed as a whole but also separately considering land-only and ocean-only grid-cells.

### 4.1. Overview Statistics

Table 2 provides an overview of the area weighted mean values of the different datasets for the previously defined regions (Tropics, with and without a separation of land- and ocean-only values, Pacific ITCZ, Central Europe, and the SA Monsoon region) as well as for global, global land-only, and the latitude belt 60° S–60° N for the common analysis period 1983–2016. Additionally, the table also provides the area-weighted pattern correlations with the reference dataset GPCP-SG, as well as the area-weighted average root-mean-square deviation (RMSD) from GPCP-SG. The RMSD value is calculated as square root of the average of the area-weighted squared residuals between the values of the respective dataset and GPCP-SG. Values for RMSD are always positive, and smaller

RMSD values indicate fewer deviations from the reference dataset. With these reported metrics for each of the discussed regions, Table 2 can be seen as numerical summary of the analyses and findings described in Section 4.2, Section 4.3, Section 4.4, and Section 4.5.

**Table 2.** Area weighted mean values (in mm day$^{-1}$) of the different datasets for the previously defined regions regridded to a common 2.5° × 2.5° grid and for the common analysis period 1983–2016, their respective pattern correlations (dimensionless) and root-mean-square deviation (RMSD, mm day$^{-1}$) to the dataset GPCP-SG.

| | Mean (mm day$^{-1}$) | Correlation | RMSD (mm day$^{-1}$) |
|---|---|---|---|
| **Global (all)** | | | |
| ERA5 | 2.914 | 0.925 | 0.898 |
| ERA-Interim | 2.926 | 0.919 | 0.944 |
| JRA-55 | 3.268 | 0.910 | 1.312 |
| MERRA-2 | 2.976 | 0.813 | 1.678 |
| GPCP-SG | 2.692 | 1.0 | 0.0 |
| **Global (land-only)** | | | |
| ERA5 | 2.308 | 0.854 | 1.331 |
| ERA5-Land | 2.263 | 0.854 | 1.179 |
| ERA-Interim | 2.183 | 0.842 | 0.986 |
| JRA-55 | 2.324 | 0.913 | 0.882 |
| MERRA-2 | 2.706 | 0.713 | 2.907 |
| WFDE5 | 2.125 | 0.949 | 0.657 |
| GPCP-SG | 2.182 | 1.0 | 0.0 |
| GPCC | 2.166 | 0.953 | 0.346 |
| **60° S to 60° N (all)** | | | |
| ERA5 | 3.147 | 0.920 | 0.949 |
| ERA-Interim | 3.180 | 0.913 | 1.004 |
| JRA-55 | 3.547 | 0.903 | 1.400 |
| MERRA-2 | 3.195 | 0.804 | 1.793 |
| GPCP-SG | 2.902 | 1.0 | 0.0 |
| PERSIANN-CDR | 2.849 | 0.996 | 0.187 |
| **Tropics (all)** | | | |
| ERA5 | 3.453 | 0.924 | 1.163 |
| ERA-Interim | 3.666 | 0.928 | 1.244 |
| JRA-55 | 4.111 | 0.916 | 1.796 |
| MERRA-2 | 3.589 | 0.800 | 2.313 |
| GPCP-SG | 3.059 | 1.0 | 0.0 |
| PERSIANN-CDR | 2.995 | 0.996 | 0.226 |
| **Tropics (land-only)** | | | |
| ERA5 | 3.246 | 0.833 | 1.852 |
| ERA5-Land | 3.181 | 0.845 | 1.618 |
| ERA-Interim | 3.167 | 0.841 | 1.356 |
| JRA-55 | 3.235 | 0.899 | 1.196 |
| MERRA-2 | 4.014 | 0.688 | 4.268 |
| WFDE5 | 3.034 | 0.941 | 0.875 |
| GPCP-SG | 3.066 | 1.0 | 0.0 |
| GPCC | 2.968 | 0.989 | 0.366 |
| PERSIANN-CDR | 2.940 | 0.955 | 0.299 |
| **Tropics (ocean-only)** | | | |
| ERA5 | 3.540 | 0.971 | 0.809 |
| ERA-Interim | 3.797 | 0.927 | 1.215 |
| JRA-55 | 4.424 | 0.943 | 1.961 |
| MERRA-2 | 3.396 | 0.940 | 0.938 |
| GPCP-SG | 3.070 | 1.0 | 0.0 |
| PERSIANN-CDR | 2.982 | 0.987 | 0.185 |

**Table 2.** *Cont.*

|  | Mean (mm day$^{-1}$) | Correlation | RMSD (mm day$^{-1}$) |
|---|---|---|---|
| **Pacific ITCZ** | | | |
| ERA5 | 6.564 | 0.966 | 1.287 |
| ERA-Interim | 6.878 | 0.972 | 1.472 |
| JRA-55 | 8.445 | 0.934 | 3.197 |
| MERRA-2 | 5.993 | 0.976 | 0.696 |
| GPCP-SG | 5.570 | 1.0 | 0.0 |
| PERSIANN-CDR | 5.542 | 0.999 | 0.179 |
| **Central Europe (land-only)** | | | |
| ERA5 | 2.593 | 0.862 | 0.434 |
| ERA5-Land | 2.616 | 0.875 | 0.416 |
| ERA-Interim | 2.290 | 0.801 | 0.553 |
| JRA-55 | 2.472 | 0.817 | 0.425 |
| MERRA-2 | 2.649 | 0.810 | 0.474 |
| WFDE5 | 2.619 | 0.823 | 0.655 |
| GPCP-SG | 2.756 | 1.0 | 0.0 |
| GPCC | 2.399 | 0.971 | 0.382 |
| PERSIANN-CDR | 2.792 | 0.970 | 0.082 |
| E-OBS | 2.245 | 0.792 | 0.803 |
| **SA Monsoon** | | | |
| ERA5 | 3.876 | 0.902 | 0.994 |
| ERA-Interim | 4.090 | 0.753 | 1.638 |
| JRA-55 | 5.153 | 0.808 | 2.168 |
| MERRA-2 | 4.359 | 0.717 | 1.823 |
| GPCP-SG | 3.646 | 1.0 | 0.0 |
| PERSIANN-CDR | 3.648 | 0.990 | 0.250 |

*4.2. Geographical Distribution of Precipitation Rate Climatologies*

Nine datasets were used in the comparison of global maps of precipitation rate climatologies: five reanalysis datasets (ERA5, ERA5-Land, ERA-Interim, JRA-55, and MERRA-2), three observational datasets (GPCP-SG, GPCC, and PERSIANN-CDR), and one bias-corrected reanalysis (WFDE5). The common time period covered by the nine datasets is 1983 to 2016. Similar to [34], GPCP-SG was used as reference dataset when calculating the differences in precipitation among the different datasets as this observational dataset is widely used as a reference dataset for precipitation (e.g., [5,75,76]). E-OBS and TRMM-L3 were not included in this analysis since they only cover limited regions of the global land surface or their temporal coverage is shorter than for the other datasets, respectively.

The geographical distributions of the multi-year annual mean climatologies of precipitation rates from all datasets calculated over the time period 1983–2016 are shown in Figure 2. For this analysis, the datasets were plotted in their native spatial resolution. This leads to a more detailed depiction of the features of the precipitation rate visible in the datasets ERA5, ERA5-Land, ERA-Interim, WFDE5, JRA-55, MERRA-2, GPCC, and PERSIANN-CDR compared to GPCP-SG. Typical features of the well-known global precipitation patterns are clearly present in all datasets: local rainfall maxima in the ITCZ stretching in the Tropics from the West Pacific over South America to the Atlantic, the tropical warm-pool region, the North Pacific and the North Atlantic regions just off the coastlines of the USA and Japan, and the South Pacific Convergence Zone (SPCZ). These patterns are consistent with geographical patterns that have been shown before (e.g., [2,15]).

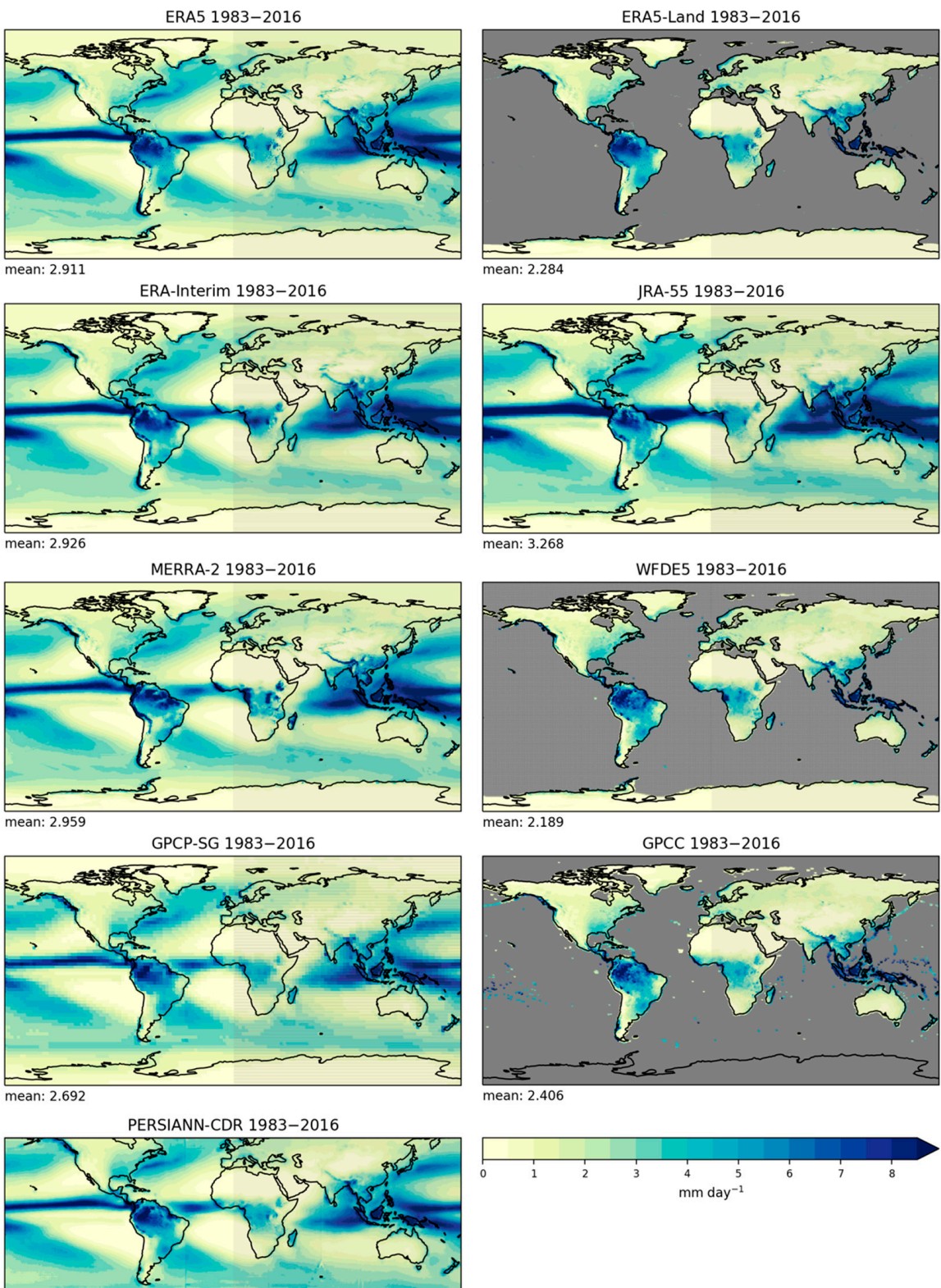

**Figure 2.** Global multi-year annual mean precipitation rates averaged over the time period 1983–2016 from the nine global datasets covering this time period. Data are shown in their native spatial resolution.

Table 2 shows the climatological mean values for the precipitation rates averaged over the three regions (1) global (land + ocean), (2) global (land-only), and (3) 60° S to 60° N (land + ocean) and the corresponding values of linear pattern correlations with respect to the observational product GPCP-SG used as a reference dataset. It can be seen in Table 2 that the overall mean climatological values for global and near-global areas are higher for the reanalysis datasets (ERA5, ERA5-Land, ERA-Interim, JRA-55, and MERRA-2) than for the observations (GPCP-SG, GPCC, and PERSIANN-CDR), with JRA-55 as the highest in the category where land and ocean values are both considered, and MERRA-2 as the highest in the category where land-only values are considered.

The pattern or spatial correlations are a measure to describe how well a dataset represents the geographical distribution of precipitation rate values. The closer the value for the spatial correlation is to 1, the better the respective dataset reproduces the spatial distribution of the chosen "standard" dataset (here: GPCP-SG). In general, the spatial correlations between the different datasets and GPCP-SG are high (>0.85), especially if global data with both land and ocean values are considered. The exception is the correlation between GPCP-SG with MERRA-2 that does not exceed 0.81 in all three analyzed global and near-global categories. Additionally, it becomes obvious that the three purely observational datasets are very similar (GPCP-SG, GPCC, and PERSIANN-CDR; correlation coefficient for PERSIANN-CDF for the region 60° S to 60° N > 0.99 and for GPCC for global land-only >0.95). The WFDE5 (bias-corrected reanalysis) dataset shows that the overall values for the climatology and correlation are more similar to the observations than the other reanalysis datasets. Overall, the different datasets represent the geographical precipitation rate patterns well, and the spread between the climatological mean values is rather small.

When looking at the spatial biases between the five reanalysis datasets and the observational product GPCP-SG (see Figure 3) features of well-known problems become apparent: there is a wet bias over the tropical and subtropical oceans, the Arctic Ocean, as well as over Central Africa and the Indian Ocean. Additionally, there is a dry bias over parts of the Northern Hemisphere (NH) continental areas, the northern part of Africa, and over Antarctica. It also seems that the reanalysis products show an overestimation of precipitation rates in high mountain ranges such as the Andes and Himalayas. Interestingly, the wet bias over Central Africa and the Indian Ocean, and the dry bias over the NH continental areas are notably reduced in ERA5 (typically <0.3 mm day$^{-1}$) compared to ERA-Interim (typically <0.5 mm day$^{-1}$). MERRA-2 shows also a relatively small dry bias over the NH continental areas of typically <0.5 mm day$^{-1}$ but shows a very pronounced wet bias in the northern high latitudes and parts of eastern Siberia of up to about 1 mm day$^{-1}$. JRA-55 shows stronger wet biases of up to about 5 mm day$^{-1}$ in the areas of deep convection in the Tropics compared to the other reanalyses but has less pronounced dry biases. It is also clear from Figure 3 (and Table 2) that two of the three observational datasets (GPCP-SG and PERSIANN-CDR) are very similar. There are only very few regions where PERSIANN-CDR shows differences compared with GPCP-SG, and these differences are small overall (typically <0.3 mm day$^{-1}$). This is not surprising since PERSIANN-CDR was bias corrected with GPCP data. WFDE5 as a bias-corrected reanalysis dataset also shows overall relatively small biases of typically <0.5 mm day$^{-1}$ compared to GPCP-SG and is therefore more in-line with the purely observational datasets (PERSIANN-CDR and GPCC). A pronounced pattern of alternating biases in Africa is found in WFDE5 and GPCC that is not present in other datasets. The similarities between WFDE5 and GPCC are not surprising since WFDE5 was bias-corrected on the basis of GPCC and CRU data. WFDE5 offsets are the largest in data sparse terrestrial regions such as Antarctica, pointing to the potential limitations in the bias-correction approach employed in that product.

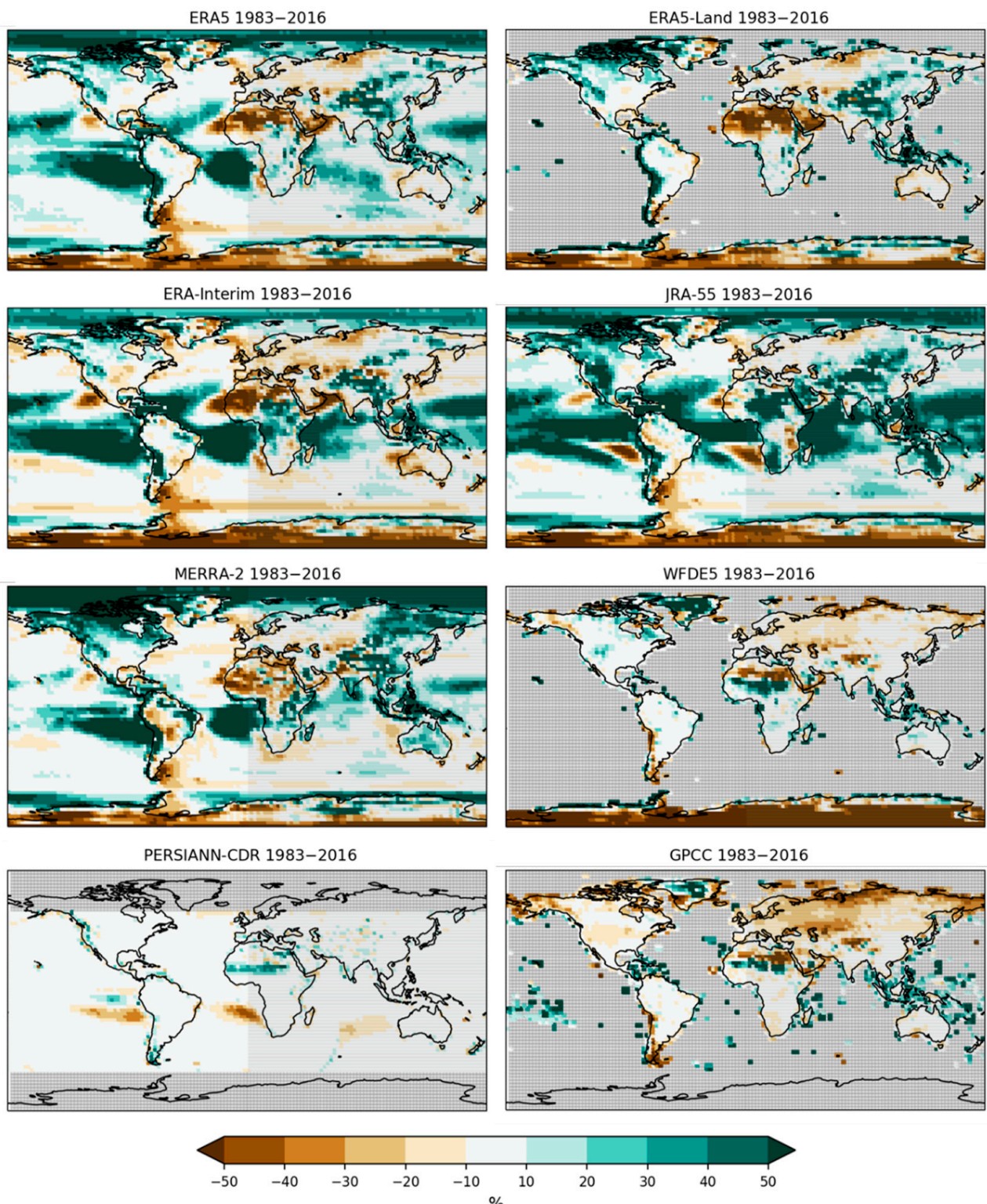

**Figure 3.** Relative differences (bias) of the multi-year annual means for eight datasets in % with respect to the observational dataset GPCP-SG (reference dataset), averaged over the time period 1983–2016. Green colors correspond to a wet bias, brown colors to a dry bias. Gray grid cells indicate no data.

*4.3. Histograms of Precipitation Rate Values*

In addition to comparing multi-year climatologies, biases, and time series, it is helpful to analyze regions with specific precipitation regimes separately. The regions that were selected for this analysis and their specific precipitation characteristics are described in Section 3.2 and shown in Figure 1.

Figure 4 shows the frequency of occurrence for the monthly mean precipitation rates ranging from 0 to 12 mm day$^{-1}$ (left column) and 0 to 1 mm day$^{-1}$ (right column) in the selected and analyzed regions. For the calculation of the histograms shown in Figure 4, all datasets have been regridded to the $2.5° \times 2.5°$ grid of the GPCP-SG dataset, and precipitation values were sampled from all months for the period 1983–2016 and grid cells in the given region. The precipitation rates in the Tropics show in all datasets a distribution that is strongly skewed towards very low values < 0.1 mm day$^{-1}$, with higher values clearly occurring less frequently (Figure 4, left column). As illustrated in the top right panel of Figure 4, the lowest monthly mean precipitation values are most frequent in the observationally-based datasets (GPCP-SG and PERSIANN-CDR) and JRA-55 which has been shown in Section 4.2 to have the highest global mean value of all datasets analyzed here. The low precipitation rates from JRA-55 in the Tropics point therefore to an overestimation of precipitation in other regions resulting in an overestimation of the global mean. When zooming in on the lower value histogram bins, it becomes clear that very small monthly mean precipitation rates of less than 0.2 mm day$^{-1}$ are underestimated in ERA5, ERA-Interim, and MERRA-2 compared to the observations, and precipitation rates > 0.5 mm day$^{-1}$ are frequently overestimated.

When the precipitation rate frequency distribution is analyzed for land-only grid cells (Figure 4, middle row), it is apparent that these values are similarly distributed with the smaller precipitation rates being the most frequent. The differences are small between the observational datasets and ERA5, ERA5-Land, ERA-Interim, WFDE5, and MERRA-2.

When looking at the distribution of the precipitation rates over ocean grid cells, the histograms reveal that the values of the lowest bin have very similar frequencies, with the observational datasets and JRA-55 being between 5 and 15% more frequent than ERA5, ERA-Interim, and MERRA-2. Zooming in on the lowest precipitation rate bin, it becomes clear that small precipitation rates of up to 0.25 mm day$^{-1}$ are underestimated by ERA5 and ERA-Interim when compared to the observations, similar to the overall tropical precipitation rate distributions. However, for the ocean-only values, the GPCP-SG frequencies are about 100% higher than the frequencies of ERA5 and ERA-Interim, and frequencies of JRA-55 and PERSIANN-CDR can be up to 200% higher. MERRA-2 shows the lowest bin frequencies similar to ERA5/ERA-Interim, therefore underestimating the occurrence of low precipitation rates compared to the observations. However, regarding the frequencies in the next larger bins of low precipitation rates, MERRA-2 shows the highest values of all datasets for ocean-only values, again similar to the distribution of ERA5/ERA-Interim. These similarities and differences in the land-only and ocean frequencies, respectively, point towards a systematic bias in the ERA5/ERA-Interim and MERRA-2 datasets over tropical oceans assuming that the observations show a realistic frequency distribution. Similar findings have been reported in a tropical cyclone study [77]. Here, more detailed analyses of daily and sub-daily values would be needed to better understand the possible reasons for this.

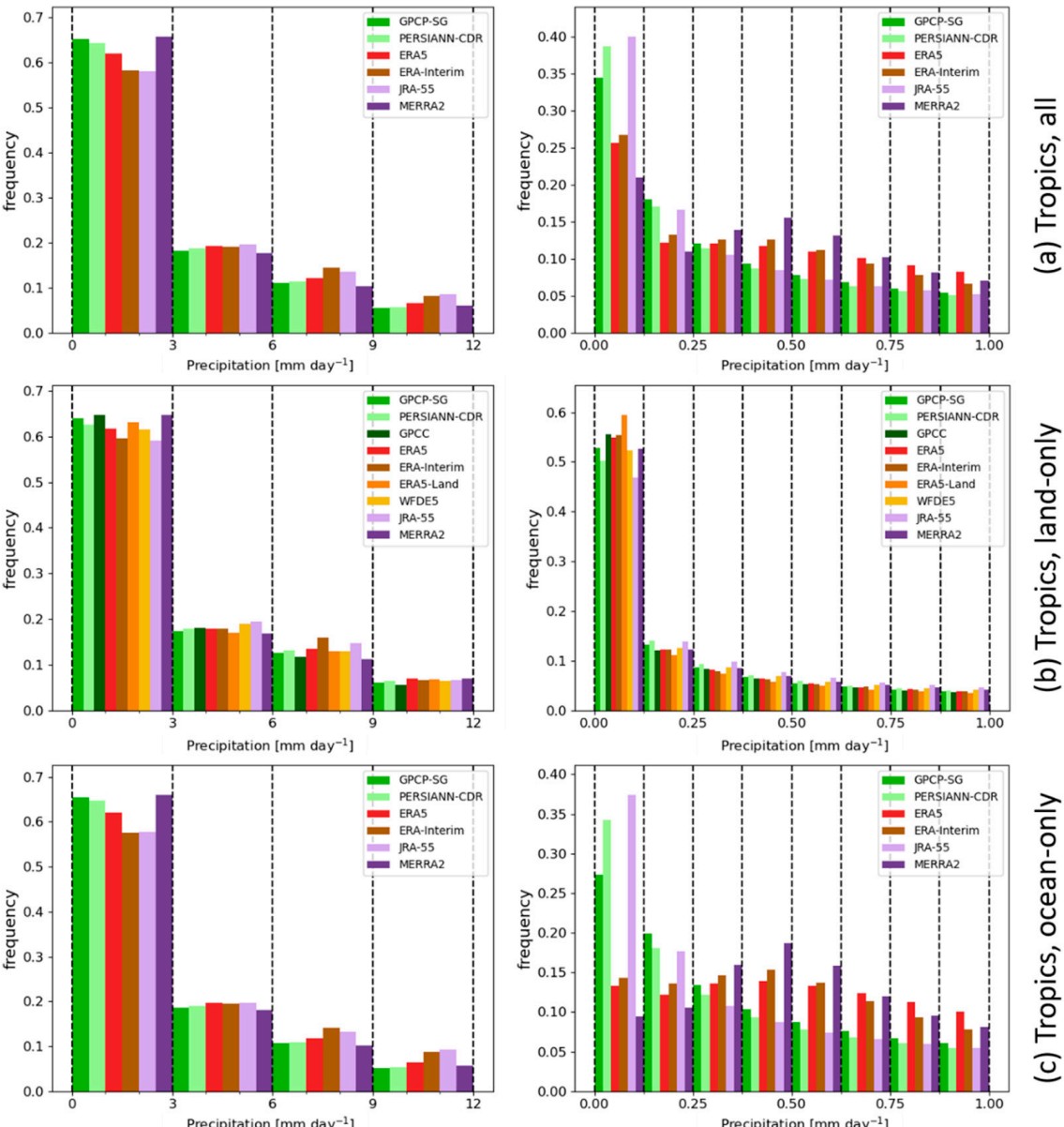

**Figure 4.** Histograms of precipitation rates in the Tropics (30° S–30° N) with no values masked (**a**), ocean values (**b**) or land values masked (**c**), for the 0–12 mm day$^{-1}$ range (left column) and 0–1 mm day$^{-1}$ range (right column). Histograms were calculated using monthly values in the period 1983–2016. ERA5-Land, GPCC, and WFDE5 are available over land only.

Interestingly, when analyzing the precipitation rate distribution in the Pacific ITCZ region (Figure 5, upper row), a region within the tropics with a frequent occurrence of deep convection, the distributions of the four reanalysis datasets differ substantially. ERA5 and MERRA-2 show a distribution closer to GPCP-SG and PERSIANN-CDR, with higher frequencies in the lowest precipitation rate bins and roughly similar frequencies in the range from approx. 3 to 9 mm day$^{-1}$. ERA-Interim underestimates the frequency of the lowest bins, then overestimates the frequencies in the higher bins of the histograms. Surprisingly, and in contrast to the distribution for the entire Tropics (Figure 4, top right panel), JRA-55 clearly underestimates the lowest precipitation rate frequencies except the lowest bin shown in Figure 5 (top right panel), and overestimates the precipitation rates in the higher bins. The contribution of the 0–3 mm day$^{-1}$ bin to the total precipitation in the Tropics in JRA-55 is about 15%. The overestimation of the frequencies in the higher bins is therefore the main reason why JRA-55 exhibits a high bias in the average precipitation rate compared to the observations.

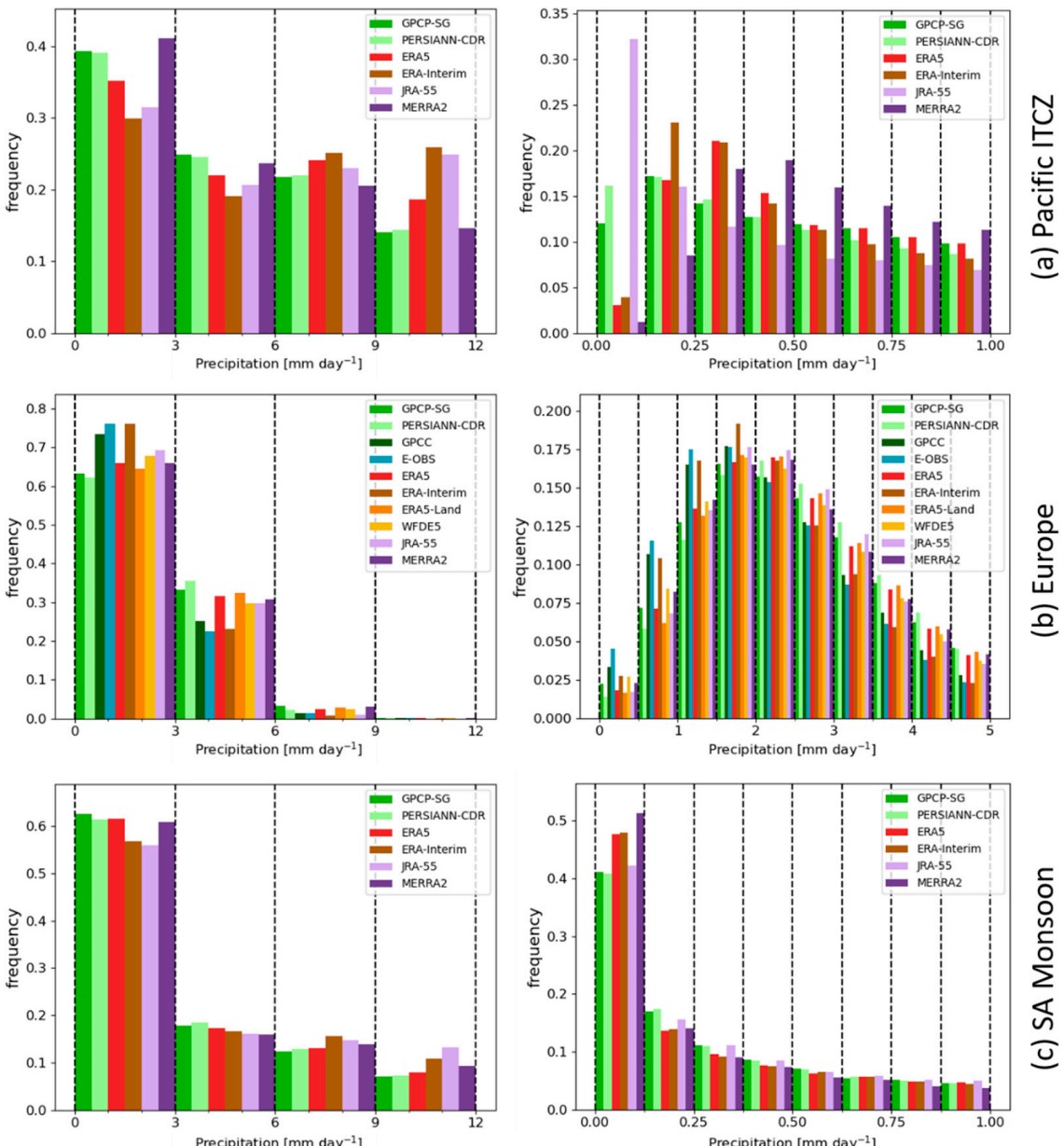

**Figure 5.** Histograms of the precipitation rate values in (**a**) the Pacific ITCZ (0°–12° N, 136° E–85° W), (**b**) Central Europe (42° N–53° N, 0°–20° E), and (**c**) the SA Monsoon region (5° N–30° N, 65° E–95° E) with no values masked, for the 0–12 mm day$^{-1}$ range (left column) and 0–1 mm day$^{-1}$ range for the Pacific ITCZ and SA Monsoon regions and 0–5 mm day$^{-1}$ for Europe (right column). Histograms were calculated using monthly mean values in the period 1983–2016. ERA5-Land, GPCC, and WFDE5 are only shown in the histograms for regions without ocean grid cells.

The precipitation frequencies for Central Europe (Figure 5, middle row) show for all datasets a distinct peak in the frequency of the precipitation rates at around 1.5–2.5 mm day$^{-1}$. However, ERA-Interim, GPCC, and E-OBS have their peak shifted a little more towards smaller values (1.5–2 mm day$^{-1}$) whereas JRA-55 and MERRA-2 show the peak a little more towards higher values (2–2.5 mm day$^{-1}$). ERA5, ERA5-Land, and WFDE5 follow the distribution of GPCP-SG and PERSIANN-CDR relatively closely, both in terms of frequencies and shape.

The distribution of the SA Monsoon region is again dominated by the frequencies in the lowest histogram bins (a pattern similar to the distributions for the entire Tropics), and all datasets exhibit very similar distributions. Even when zooming into the smallest histogram bin, the distribution of the datasets is very similar, with only ERA5, ERA-interim,

and MERRA-2 showing slightly higher frequencies in the lowest bin (<0.1 mm day$^{-1}$) by comparison with the observations.

*4.4. Monthly Mean Area Averaged Time Series of Precipitation Rates*

Monthly mean area averaged time series Figures 6–8) provide an overview of the temporal evolution of precipitation rates in the analyzed regions. For each month in the period 1983 to 2016 the average precipitation was calculated as an area-weighted mean for each of the analyzed datasets. In order to allow the inclusion of the additional observational dataset TRMM-L3 which has a shorter temporal coverage than the other datasets, the monthly mean area average time series in the Tropics was also calculated for the time period 1998 to 2013 (Figure 7).

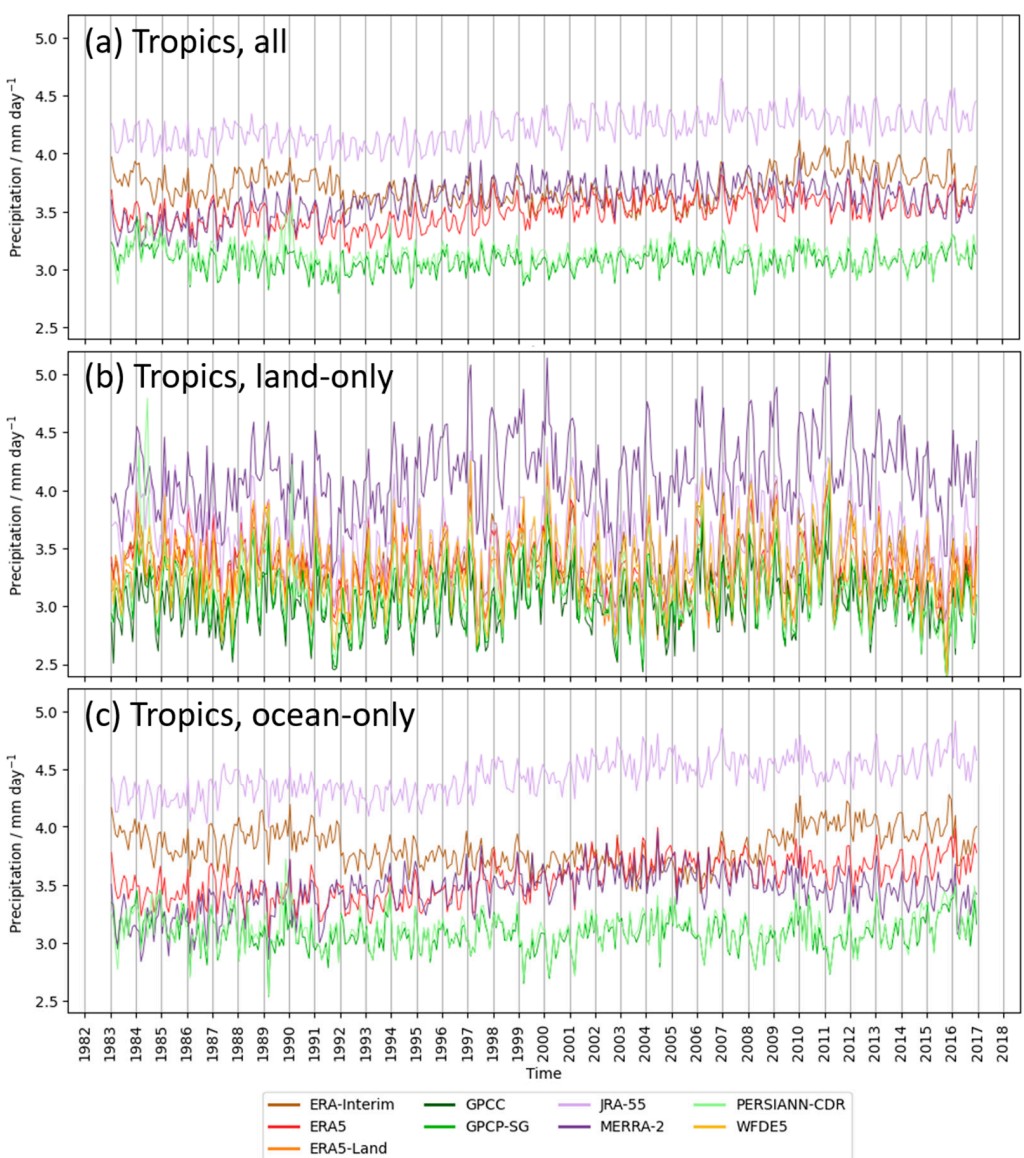

**Figure 6.** Time series of monthly mean precipitation rates averaged over the Tropics (30° S–30° N) for the period 1983–2016. (**a**) entire tropical region; (**b**) land-only values in the tropical region (including ERA5-Land, GPCC and WFDE5 data); (**c**) ocean-only values in the tropical region.

Figure 6 shows the monthly mean time series averaged over the whole Tropics (land and ocean (a)), the land-only values (b), and the ocean-only values (c). The observational datasets GPCP-SG and PERSIANN-CDR are very similar in all three cases with almost identical precipitation rates (between 2.8 and 3.3 mm day$^{-1}$ for the whole Tropics) and

variability. For the whole Tropics (Figure 6a) the reanalysis datasets exhibit distinctly higher precipitation rates (between 3.3 and 4.5 mm day$^{-1}$ over the course of the analyzed time period), with JRA-55 showing the highest value of all reanalyses being up to 35% higher than the observations. The precipitation rates for ERA5 and ERA-Interim are very similar in magnitude between about 1998 and 2007 but they diverge considerably before and after that time period with ERA-Interim being up to 15% higher. ERA5 shows a weaker long-term variability with respect to its predecessor ERA-Interim which may be attributable to several years of model and data assimilation developments [16]. MERRA-2 shows values similar to ERA5 during most of the time period analyzed; only in the early 1990s are the differences more pronounced (up to 0.25 mm day$^{-1}$).

An analysis of tropical land-only values shows that the differences between the datasets in the monthly mean tropical average time series values are clearly reduced. The spread between most datasets with respect to GPCP-SG is roughly between 10 and 25% with the exception of MERRA-2 that can be up to 50% higher than the other datasets. The spread is basically determined by a systematic offset with the short- and long-term variability being very similar in all dataset time series. Considering the land-only values, the observational datasets seem to be on the lower end of the spread across the different datasets (green lines in Figure 6b, between about 2.8 and 3.4 mm day$^{-1}$), whereas the MERRA-2 values are notably the highest among all datasets (between about 4.2 and 4.7 mm day$^{-1}$). It is also noteworthy that the observational datasets for the whole Tropics and the land-only tropical values are of almost of the same magnitude, i.e., in the mean approx. 3 mm day$^{-1}$. This is in very good agreement with the mean value for the Tropics (around 3 mm day$^{-1}$ for both GPCP-SG and PERSIANN-CDR) and a little higher than the 60° S to 60° N value presented in Table 2 (around 2.9 mm day$^{-1}$ for both GPCP-SG and PERSIANN-CDR). While there is almost no difference between the observationally based data products averaged over the whole Tropics and the land-only tropical region, the reanalyses show a clear difference, with the exception of MERRA-2: the overall tropical values are mostly higher than the land-only values (3.3 to 4.5 mm day$^{-1}$ and 3.3 to 3.7 mm day$^{-1}$, respectively). This was further investigated by looking also at ocean-only values (Figure 6c) that are also higher in MERRA-2 than in the observations. For MERRA-2, the values over land in the Tropics are slightly higher than over oceans or the entire Tropics (up to 1 mm day$^{-1}$). Overall, the results indicate again the presence of large systematic differences between the observational and reanalysis data time series for the whole tropical region. These originate primarily from the precipitation values over the ocean (except for MERRA-2), which might be a result of more observational data being available for data assimilation over land.

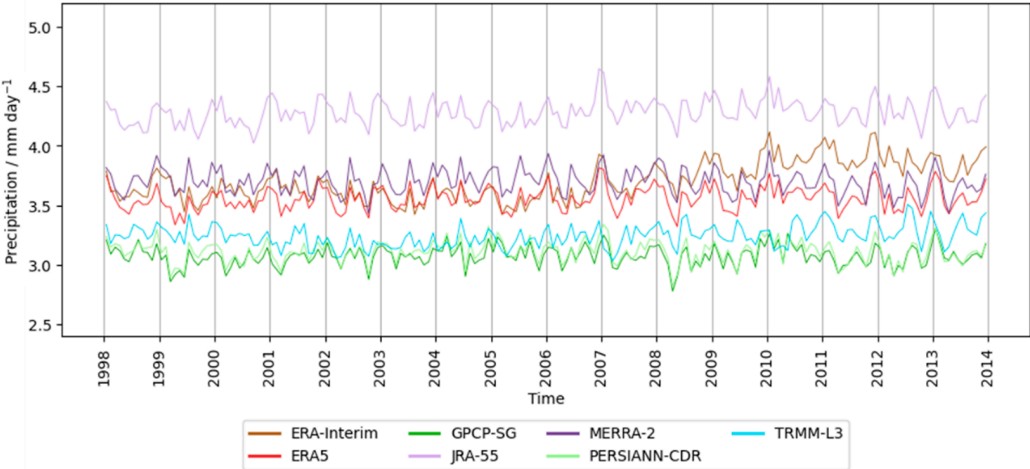

**Figure 7.** Same as Figure 6a but for the shorter time period 1998–2013 for which TRMM-L3 data are available.

For a better understanding of the differences between the observations and reanalyses in the Tropics, the Tropical Rainfall Measuring Mission (TRMM-L3) was used as an additional independent observational dataset. This dataset is only available for the Tropics and the subtropics (50° S–50° N), and the version available from obs4MIPs covers only the time period 1998 to 2013. This is the reason why TRMM-L3 was not used in other analyses presented here. Figure 7 shows the monthly mean precipitation rates averaged over the whole Tropics for the shorter time period 1998 to 2013. The TRMM-L3 dataset shows values considerably closer to the other two observational datasets than any of the reanalyses (with a positive bias of up to 0.3 mm day$^{-1}$) supporting the earlier findings that all reanalysis datasets analyzed here overestimate the monthly mean precipitation rates over some parts of the tropical oceans (or the tropical land masses in the case of MERRA-2) to some degree compared to observations.

The monthly mean precipitation rates in the Pacific ITCZ (Figure 8a) show in all datasets a higher consistency with each other (spread smaller than 1.5 mm day$^{-1}$ in most months), than is the case for the ocean-only tropical precipitation rate values shown in Figure 6 (spread in values about 1.5 mm day$^{-1}$). However, there is still a spread between the datasets with a clustering of the observational datasets towards the lower end, and JRA-55 at the high end of precipitation rates (up to 12 mm day$^{-1}$ in selected months). The overall interannual variability for all the datasets is very similar, and the clear signal of the extreme El Niño event in 1997/1998 (e.g., [78]) and a second very intense event in 2015/2016 (e.g., [20]) is apparent in all time series.

The time series of the monthly mean precipitation rates averaged over Central Europe (Figure 8b) show almost no differences between the reanalysis and observational datasets (spread mostly <0.3 mm day$^{-1}$). The observed precipitation rates and also the interannual variability are very well reproduced by the reanalyses (mean, correlation, and RSMD values are very similar for all datasets, see also Section 4.1 and Table 2), such as the strong precipitation anomaly related to the exceptionally hot summer in 2003 in Europe [79,80]. This high similarity between the reanalyses and observations might have been made possible by the availability of numerous high-quality ground-based measurements that could be assimilated in the production of the reanalyses. The similarly well reproduced observations over the continental U.S. (not shown here) support this suggestion.

The time series of the monthly mean precipitation rates averaged over the SA Monsoon region (Figure 8c) are dominated by a distinct annual cycle with an amplitude of up to 10 mm day$^{-1}$. All datasets show this annual cycle with a peak in boreal summer (monsoon season) and a minimum during boreal wintertime (dry season). The two observational datasets and the reanalyses ERA5 and ERA-Interim show very similar values, with the reanalyses slightly overestimating the maximum values of the average annual cycle (mostly <1 mm day$^{-1}$). JRA-55, and to some extent also MERRA-2, show clearly higher maximum values each year (up to 2 mm day$^{-1}$), which is consistent with their overall higher mean values (Table 2) and the biases described above and shown in Figure 3.

Monthly mean area-averaged time series of precipitation anomalies are presented in Figure 9. These were derived as monthly anomalies with respect to the climatology of the whole time series (1983–2016) for each individual dataset. For instance, the anomalies for ERA5 were calculated with respect to the ERA5 long-term climatology. The anomalies are presented as a function of time and were normalized to show the deviation from the mean as a multiple of the standard deviation of each individual dataset. The anomaly plots provide a good overview of the temporal evolution for the precipitation rate time series, both for short- and long-term fluctuations, and also show how well the individual datasets represent well-known deviations from the mean (very dry or very wet months). Datasets that have missing values within the region of interest (e.g., PERSIANN-CDR) were excluded from this analysis as a distinction between land-only and ocean-only data was not done for this analysis.

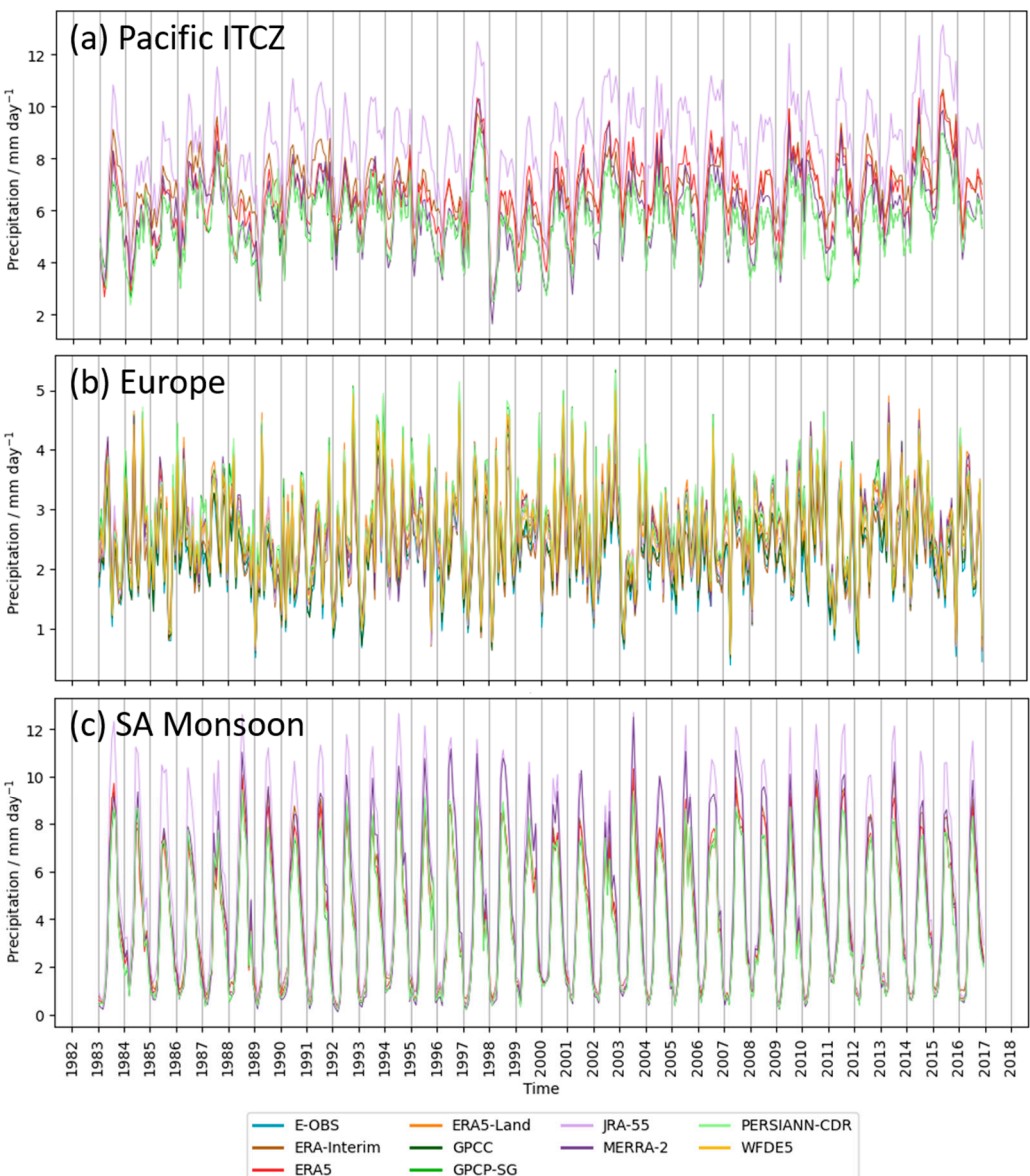

**Figure 8.** Times series of area averaged monthly mean precipitation rates for (**a**) the Pacific ITCZ, (**b**) Central Europe (including ERA5-Land, GPCC and WFDE5 data), and the (**c**) SA Monsoon region for the period 1983–2016.

Figure 9 shows the anomaly time series for three regions defined in Section 3.2, the entire Tropics (a), the Pacific ITCZ (b) and Central Europe (c). As already seen in the analyses presented above, the anomaly time series for the Tropics for the different datasets are markedly different for this region. The normalized anomalies can vary by up to a factor of two (i.e., two standard deviations), and for some periods show different signs. In the period 2002 to 2007, ERA-Interim shows negative anomalies, whereas ERA5 and JRA-55, and also to some extent GPCP-SG, show positive anomalies. This spread in the anomalies between the different time series can also be seen in the monthly mean time series (Figure 6).

Anomaly time series for the Pacific ITCZ region and Central Europe are much more consistent across the different datasets. In most cases the anomalies from all five (for Pacific ITCZ) or nine (for Central Europe) datasets are almost indistinguishable in Figure 9b,c. They also indicate anomalies of similar magnitude for well-known climatic fluctuations such as, for instance, the effects of the strong El Niño event in 1997/1998 (e.g., [78]) or 2015/2016 [20], or the exceptionally hot and dry summer in Europe in 2003 (e.g., [79]) and 2015 [81]. For these regions all reanalysis datasets describe the temporal evolution of

precipitation rates and its variability well (differences are mostly <0.1 standard deviations for Europe and <0.3 standard deviations for Pacific ITCZ).

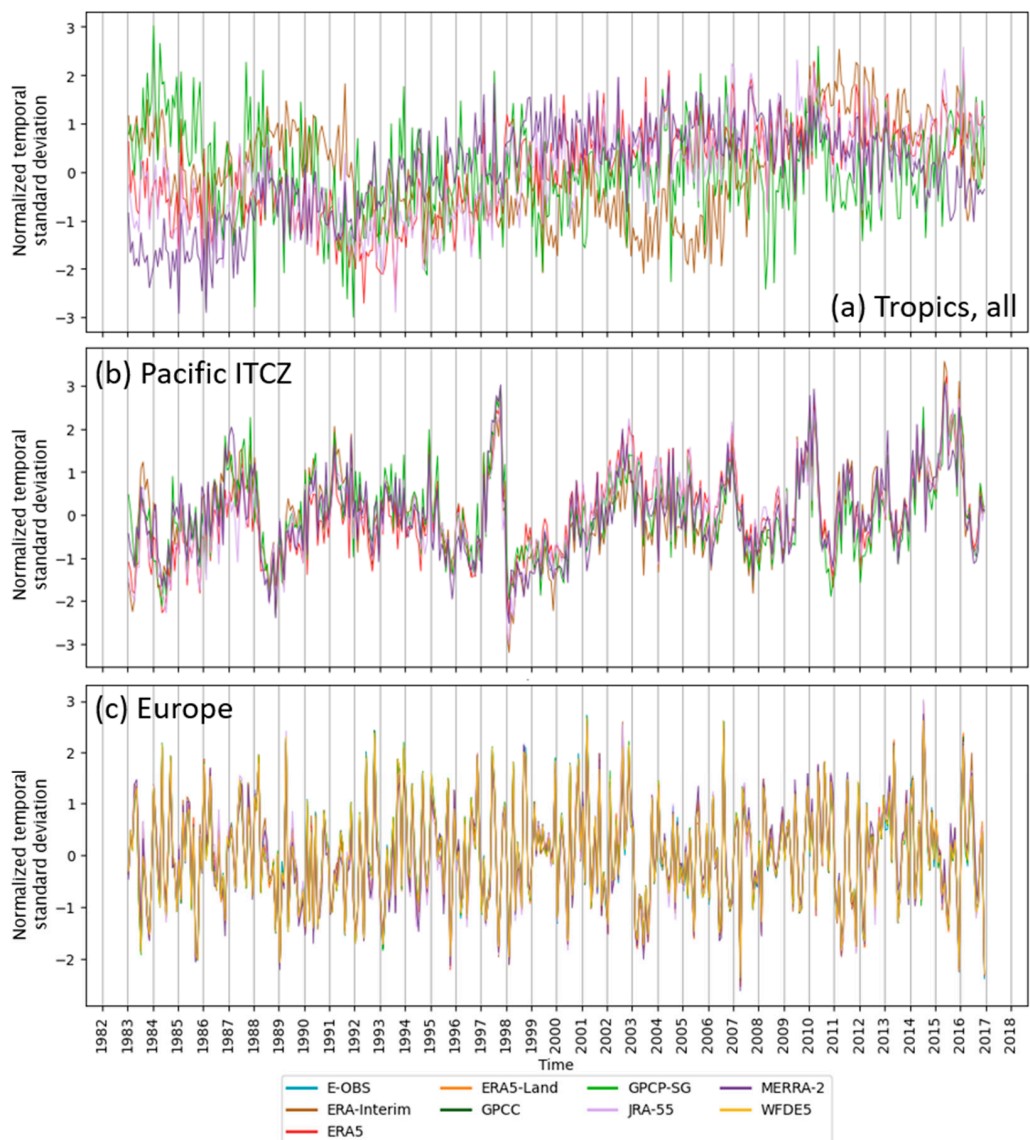

**Figure 9.** Times series of area-averaged monthly mean anomalies of precipitation rates for (**a**) the entire Tropics, (**b**) the Pacific ITCZ and (**c**) Central Europe calculated with respect to the respective dataset climatology for the period 1983–2016. Anomalies are shown in multiples of the temporal standard deviation.

*4.5. Annual Cycle of Precipitation Rates*

Climatological annual cycles of all datasets calculated over the full analyzed period 1983–2016 are provided in Figures 10 and 11. With such comparisons, it is possible to better assess the biases between datasets that are caused by the underlying climatology. They also help to determine whether biases have a specific seasonal component, or whether they are roughly constant throughout the year, and to determine if the shape of the annual cycle is similar among the different datasets. Similar to the analyses presented in Sections 4.3 and 4.4, an analysis was performed for the whole Tropics and for land-only and ocean grid cells (Figure 10), and additionally for the Pacific ITCZ, Central Europe and SA Monsoon regions (Figure 11).

The annual cycle of precipitation rates in the Tropics averaged over the latitude belt 30° S to 30° N is characterized by almost no variation throughout the year. The annual

cycle of the four reanalysis datasets compare well to the two observational datasets in this respect (annual cycle amplitude of <0.2 mm day$^{-1}$, Figure 10a). However, it is again clear that the reanalyses have a bias compared to the observations. While the annual cycle of the observations shows values between 3 mm day$^{-1}$ and 3.2 mm day$^{-1}$, ERA5, ERA-Interim and MERRA-2 show values that are about 15%, 20%, or 18% higher, respectively. JRA-55 shows the highest values for the annual cycle, almost 4.3 mm day$^{-1}$, which represents an overestimation of about 36% compared to the observations.

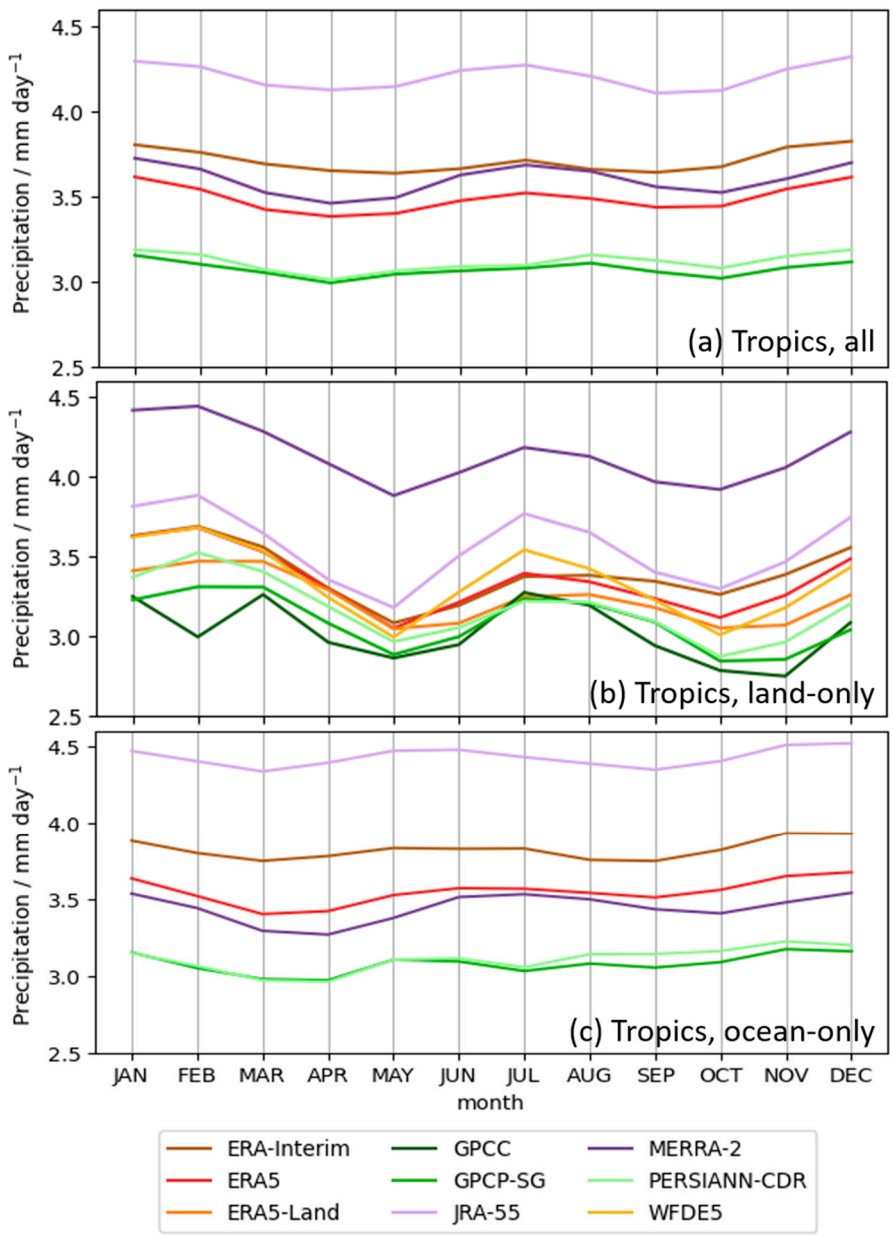

**Figure 10.** Annual cycle of precipitation rates for the Tropics calculated over the period 1983–2016. (**a**) entire Tropics; (**b**) land-only values (including ERA5-Land, GPCC and WFDE5 data); (**c**) ocean-only values.

By distinguishing between land-only and ocean grid cells when calculating the annual cycle (Figures 10b and 10c, respectively), it is clear that the overall bias in the tropical annual cycle is caused by deviations in the values over both land and ocean. In both cases, biases between the observations are of similar magnitude. The annual cycle for tropical values over land show a clear double peak in all datasets, with one maximum around February

and another around July. The biases of three of the four reanalysis datasets are similar as for the whole Tropics: the bias for JRA-55 is the highest, followed by the one for ERA-Interim, and the ERA5 bias is slightly lower than for ERA-Interim. The biggest differences between the ERA5 and ERA-Interim are found from August to December. A clear outlier for the reanalysis datasets is MERRA-2 with values of the annual cycle up to 34% higher than the observations, although values for the full Tropics are only about 18% higher than the observations. Surprisingly, the observational datasets, GPCP-SG and PERSIANN-CDR also show differences in the magnitude of their annual cycle; this is despite these two datasets showing almost identical results in the previous analyses. Differences between the datasets are most pronounced in the months November to May and can reach a magnitude of up to approx. 0.5 mm day$^{-1}$. GPCC shows the lowest values for the annual cycle of all observations and does not show the peak in February common to all other estimates. Surprisingly, the annual cycle of the WFDE5 dataset is closer in magnitude to ERA5 and ERA5-Land than GPCC, although it was bias-corrected based with GPCC.

The annual cycles calculated from values over the ocean only show almost no variations over the course of the year, and the annual cycles from the observational datasets are almost identical. The biases between the observations and the reanalyses are similar as for the whole tropical region, with JRA-55 showing the highest bias. MERRA-2 shows biases even lower than ERA5 for the values averaged over ocean grid cells only.

Figure 11 shows the precipitation rate annual cycle for the Pacific ITCZ (a), Central Europe (b) and the SA Monsoon region (c). The annual cycle for the Pacific ITCZ shows a very strong variation over the course of a year, with a distinct maximum in June, July, and August, and a minimum in February/March. The amplitude reaches up to 4 mm day$^{-1}$ for JRA-55 and is slightly smaller in the other datasets. While ERA5 and ERA-Interim show a very similar shape of the annual cycle during the months June to October, ERA5 shows lower values than ERA-Interim throughout the rest of the year. MERRA-2's Pacific ITCZ annual cycle shows the lowest values of all analyzed reanalysis datasets, but with an amplitude that is slightly more pronounced than those from observations. There is almost no difference in the annual cycles from the two observationally based datasets. Overall, the observations show the lowest values, with MERRA-2, ERA5, and ERA-Interim having slightly higher climatological values, and JRA-55 again showing the largest bias of all four reanalyses compared to the observations.

The annual cycle of the different datasets for Central Europe shows a clear double peak with a first maximum in May/June (between 2.5 and 3.1 mm day$^{-1}$) and a second one in November (between 2.6 and 3.2 mm day$^{-1}$). The spread among the datasets for the annual cycles is relatively large, spanning about 0.8 mm day$^{-1}$. GPCP-SG and PERSIANN-CDR are again very similar, but their annual cycle values are on the higher end of the spread between datasets, showing the highest values from November to February (both between 2.7 and 3.2 mm day$^{-1}$). They are exceeded from March to June by ERA5, ERA5-Land, and MERRA-2 by about 5 and 10%, for example in April. Throughout the first half of the year, ERA-Interim, WFDE5, and E-OBS show the lowest values of all the datasets (typically between 5 and 25% lower than GPCP-SG), but in the second half of the year, ERA-Interim clearly shows the lowest values of all analyzed datasets (between about 2.2 and 2.5 mm day$^{-1}$ during August through November). The values for the annual cycle from WFDE5 are in most months closer to the values shown by ERA5 than GPCC, although the bias-correction applied to WFDE5 is based on GPCC (and CRU).

The amplitude of the annual cycle in the SA Monsoon region is the most pronounced of all the regions analyzed. The precipitation rate values range from approx. 1 to 11 mm day$^{-1}$ for JRA-55, approx. 1 to 9 mm day$^{-1}$ for MERRA-2, and from approx. 1 to 8 mm day$^{-1}$ for all other datasets. JRA-55 and MERRA-2 are the only datasets showing a distinct bias in the annual cycle, whereas ERA5 and ERA-Interim show a very similar annual cycle to the observations. The SA Monsoon region climatology is dominated by a maximum precipitation rate in the months June to September, the months where the summer monsoon rainfall connected to deep convection is triggered by a strong land/ocean temperature

contrast in this region. However, in JRA-55 the relative bias remains relatively constant throughout the year at about 40–50% and the relative bias in MERRA-2 shows a seasonal dependency with the bias ranging between −5 and about 15% during December through May and about 20–30% during June through November. This compares to a relative annual mean bias of about 20% for MERRA-2 and about 40% for JRA-55 (Table 2).

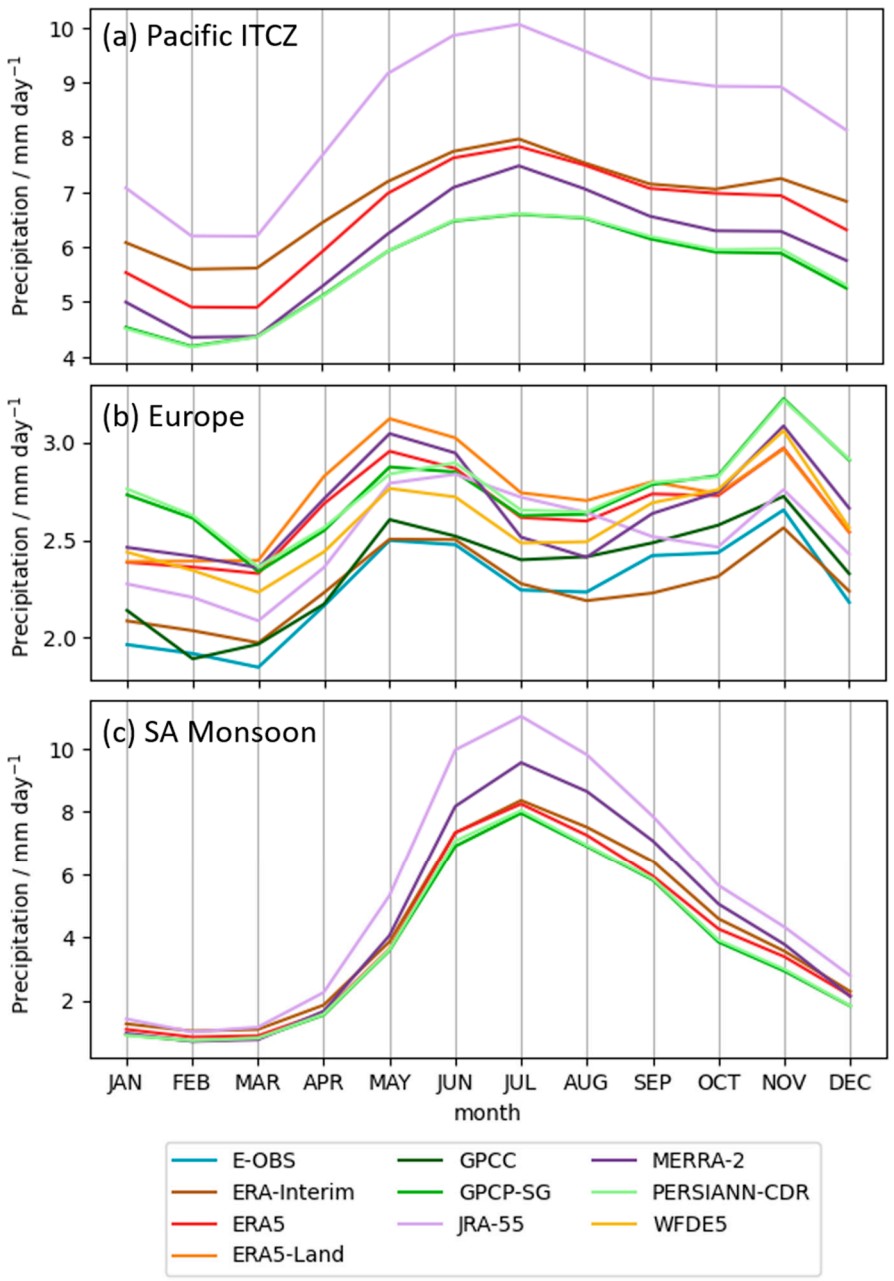

**Figure 11.** Annual cycle of precipitation rates for (**a**) the Pacific ITCZ, (**b**) Central Europe (including ERA5-Land, GPCC and WFDE5 data), and (**c**) the SA Monsoon region averaged over the period 1983–2016.

## 5. Summary and Conclusions

In this study comparisons of precipitation rates from six different reanalysis datasets including one bias-corrected reanalysis dataset and five observational datasets were presented. Overall, the global mean pattern of the precipitation rates is similar for all datasets with the most prominent features of the global precipitation climatology being present such as the ITCZ, the tropical warm-pool, the storm track regions, and the SPCZ. When

compared with the GPCP-SG dataset which is widely used as a precipitation reference dataset (e.g., [5,75,76]), the reanalyses all show well-known biases such as a wet biases over Central Africa and the Indian Ocean, the tropical and subtropical oceans, and a dry bias over the Northern Hemisphere continental areas.

Based on a range of comparisons (Sections 4.1–4.5), we conclude that most of the analyzed reanalysis datasets have difficulties in reproducing observed precipitation rate values over tropical oceans, especially in the tropical Atlantic and the tropical Indian Ocean. The exception is MERRA-2 which has problems reproducing observed precipitation rates particularly over tropical land areas. Values over land for the other reanalysis datasets compare well for area-averaged monthly mean time series and frequency distribution histograms, but slightly less well for the annual cycle climatologies. It seems therefore that the biases of most reanalysis datasets for the whole Tropics (land and ocean) are caused by differences in the precipitation rate values over the oceans, except for MERRA-2 where the differences are caused by the biases over land. However, the annual cycles of the different datasets for the Pacific ITCZ show a bias similar to the ocean-only annual cycle: JRA-55 shows the highest bias, and while ERA5, ERA-Interim, and MERRA-2 have a smaller bias, they still show higher values than the observations, and both observational datasets are very similar (which is not surprising since GPCP-SG was used to bias-correct PERSIANN-CDR).

Comparisons of the precipitation rates over Central Europe show very good agreement in all reanalyses except for the annual cycle. For this, some biases exist with ERA-Interim showing values that are too low from September to November of the annual cycle but being similar to E-OBS and GPCC during the rest of the year. GPCP-SG and PERSIANN-CDR show during half of the year the highest values of all datasets, only exceeded by ERA5-Land (and to a lesser extent ERA5 and MERRA-2) in the other half of the year. The very good agreement of the reanalyses with observations for the Central European region suggests a successful assimilation strategy of observational input data in the creation of the reanalysis products. This finding is supported by additional comparisons of the reanalyses for the continental U.S. (where many high-quality observations are available; not shown here) showing that the reanalyses are in very good agreement with observations. Maybe surprisingly, the annual cycles from WFDE5 from the tropical land areas and for Europe are closer to ERA5 values than GPCC values, although the bias correction is based on the latter dataset.

Overall, JRA-55 and MERRA-2 overestimate the precipitation rates compared to observations in many regions. ERA-Interim represents the different regions analyzed here more realistically compared to observations than JRA-55, but not as well as ERA5. The differences in the reanalysis datasets compared to observations could be caused by differences in assimilation schemes, differences in assimilated datasets, but also differences in the overall underlying model [77]. Pinning down which factor might be the most important for each reanalysis dataset is, however, very challenging in practice. Although biases still exist in ERA5, well-known problems such as the wet bias over Central Africa and the Indian Ocean and the dry bias over the Northern Hemisphere continental areas are clearly reduced compared to ERA-Interim. In addition, the frequency distribution histograms, anomaly time series, and annual cycle analyses for specific regions show an overall improvement of ERA5 compared to ERA-Interim.

The general global patterns of precipitation rates are present in all datasets and are represented well (pattern correlation between approx. 0.8 and 0.9) when compared with the observations. Observational datasets can have a limited spatial coverage (PERSIANN-CDR, E-OBS, GPCC, and TRMM-L3) and a limited temporal coverage (TRMM-L3) which can be a problem for some analyses. However, the reanalyses do show biases compared to the observations, and therefore they are also possibly not suitable for all planned analyses or applications depending on the geographical region of interest. Given the reported characteristics of the different reanalysis datasets that were discussed before, there are several details worth highlighting to potential users of these datasets:

- ERA5 and ERA5-Land represent a clear improvement over ERA-Interim based on the comparisons with the observations from GPCP-SG, PERSIANN-CDR, and TRMM-L3 (Tropics only). Given also that ERA-Interim has been discontinued, it seems good practice to use ERA5 and ERA5-Land rather than ERA-Interim for studies requiring reanalysis data;
- ERA5 and ERA5-Land show typically smaller biases in precipitation than JRA-55 and MERRA-2, especially in the Pacific ITCZ and SA Monsoon region. For the Tropics, the size of the biases differs depending on the analyzed data subset (land- or ocean-only);
- Tropical ocean precipitation rates are highly biased in three of the four reanalyses (ERA5, ERA-Interim and JRA-55), especially in the Atlantic and the Indian Ocean;
- All four reanalysis datasets with full global coverage (ERA5, ERA-Interim, JRA-55, and MERRA-2) are close to the observations over continental regions where many observations such as satellite and ground-based precipitation radar are available that can be used for assimilation in the production of the reanalysis datasets such as for Central Europe and the continental U.S;
- The bias correction on which the WFDE5 is based reduced the original ERA5 values over land but did not result in WFDE5 climatologies that were significantly closer to GPCC than ERA5;
- There are no large or fundamental differences between ERA5 and ERA5-Land due to the fact that ERA5-Land precipitation rates are derived from ERA5 by interpolation to the finer ERA5-Land grid [37].

**Author Contributions:** Both authors, B.H. and A.L., were involved in conceptualizing the study, interpreting the obtained results and writing the manuscript. B.H. performed the analyses. All authors have read and agreed to the published version of the manuscript.

**Funding:** This work was funded by European Copernicus Climate Change Service (C3S) implemented by European Centre for Medium-Range Weather Forecasts (ECMWF) under the service contract Independent Assessment on ECVs with the funding number as ECMWF/Copernicus/2017/C3S_511_CNR.

**Institutional Review Board Statement:** Not applicable.

**Informed Consent Statement:** Not applicable.

**Data Availability Statement:** All data used in this analysis are publicly available: ERA5: ERA5-Land, WFDE5, E-OBS: https://cds.climate.copernicus.eu/cdsapp#!/search?type=dataset (accessed on 2 November 2021); ERA-Interim: http://apps.ecmwf.int/datasets/data/interim-full-moda/ (accessed on 2 November 2021); MERRA-2: https://goldsmr4.gesdisc.eosdis.nasa.gov/data/MERRA2_MONTHLY/ (accessed on 2 November 2021); GPCC: https://opendata.dwd.de/climate_environment/GPCC/html/fulldata-monthly_v2018_doi_download.html (accessed on 2 November 2021); PERSIANN-CDR: https://www.ncei.noaa.gov/data/precipitation-persiann/access/ (accessed on 2 November 2021); JRA-55, GCPC-SG, TRMM-L3: https://esgf-node.llnl.gov/projects/obs4mips/ (accessed on 2 November 2021).

**Acknowledgments:** Several datasets (JRA-55, GPCP-SG, TRMM-L3) used in this work were obtained from the obs4MIPs (https://esgf-node.llnl.gov/projects/obs4mips/ (accessed on 2 November 2021)) project hosted on the Earth System Grid Federation (https://esgf.llnl.gov (accessed on 2 November 2021)). JRA-55 is provided from the Japanese 55-year Reanalysis project carried out by the Japan Meteorological Agency (JMA). The Version 2.3 GPCP-SG combined precipitation data were provided by the NCEI CDR Program as a contribution to the GEWEX Global Precipitation Climatology Project. The TRMM-L3 3B43 data were provided by the NASA/Goddard Space Flight Center's Mesoscale Atmospheric Processes Laboratory and PPS, which develop and compute the 3B43 as a contribution to TRMM. We also acknowledge the E-OBS dataset from the EU-FP6 project UERRA (http://www.uerra.eu (accessed on 2 November 2021)) and the data providers in the ECA&D project (https://www.ecad.eu (accessed on 2 November 2021)). NASA's MERRA-2 reanalysis data and PERSIANN-CDR are provided by the NCAR Climate Data Guide. GPCC is operated by Deutscher Wetterdienst (DWD, National Meteorological Service of Germany) under the auspices of the World Meteorological Organization (WMO). This manuscript contains modified Copernicus Climate Change

Service Information (2021) with the following datasets being retrieved from the Climate Data Store: ERA5, ERA5-Land, WFDE5 (neither the European Commission nor ECMWF is responsible for any use that may be made of the Copernicus Information or Data it contains). The computational resources provided by the Deutsches Klimarechenzentrum (DKRZ, Germany) were essential for performing this analysis and are kindly acknowledged.

**Conflicts of Interest:** The authors declare no conflict of interest.

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
