# Peer review of "Comparison of Reanalysis and Observational Precipitation Datasets Including ERA5 and WFDE5"

_atmosphere, doi:10.3390/atmos12111462_

Round 1

Reviewer 1 Report

General comments:

The authors explore the pros and cons of the use of different observation and reanalysis precipitation datasets in some relevant areas over the world. The paper is very interesting and of high interest for research and water management purposes. However, the design of the paper should be improved before publication. The methodology section is not adequately developed and some parts of the methods carried out are in other sections, so the paper must be reordered and rewritten. Furthermore, I find other major considerations that should be taken in account:

Introduction is lack of references related to examples of the use of datasets in other countries.

Data section should be shortened: delete historical comments, evolution, etc. (references can replace most of the text).

Discussion: Do not repeat results section.

Conclusions should be in different section

In abstract the authors indicate that GPCP is used as reference but in Line 365 refer to GPCP-SG. At least it should be validated with real precipitation gauges......no? How can you be sure that it works in all the regions???

Specific comments:

Line 17: indicate what GPCP means

Lines 17 and many others, delete “for example” and “e.g” in the most of paper

Lines 18-19 and 22: smaller and high biases-àprovide percentages of error.

Lines 21 and 52: Particularly, (add comma)

Line 36 and many others: delete “e.g.”

Line 61: “where there is little to no effective constraint-à reformulate

Lines 89: , time series of monthly mean area mean values -àreformulate

Lines 94, 99: widely used, extensively--àprovide more references

Lines 105, 364 and others: Datasets that have been used in [20]-à write the author when you refer directly as this case.

Lines 122-123: used for research purposes--àadd references

Line 147 and others: 40+ àavoid this abbreviation

Line 178, 309 and others: (last access: 1 September 2021).--àsee rules of journal for references.

Line 207: time period since 2002à add references

Line 274: provide references.

Lines 311-312: For this, the ESMValTool contains many scripts for “cmorization” of observational datasets, including-à rewrite in a more appropriate way.

Lines 336-337: Explain the analysis carried out in Methods, not in other sections.

Lines 335-369: I suggest to add a new section with the studied regions.

Figure 2 and others: why you use “climatologies” in captions?

Lines 383-384: delete “(i.e. no E-OBS and no TRMM-L3 383 data are displayed here)”

Lines 390-397: Move to Methodology section.

Line 404: “geographical distribution of low and high precipitation rate values”-à specify the differences.

Tables: Explain Correlation(1)-à indicate in caption the meaning of (1)

Lines 829-830: ….and therefore they are also possibly not suitable for 829 all planned analyses or applications.-àexplain, provide examples.

Lines 830-855: Move to new section Conclusions.

Reviewer 2 Report

In their manuscript Hassler and Lauer present a comparison of how well different gridded reanalysis datasets measure monthly rainfall through comparison with gridded observation datasets derived from weather station and satellite measurements. This includes the recently released WFDE5 and ERA5 datasets, which adds novelty to the work. Assessment of the validity and accuracy of different rainfall products in real world settings, especially outside North America and Europe, are welcome additions to the literature, and work that has the potential to have real impact.

As far as I can tell, there is nothing substantially incorrect about the work undertaken or the interpretation of the results (bar a few minor points listed below). However, the manuscript provides only a broad qualitative comparison of different products. There is little placing of results within the context of previous literature, and discussion as to what the results might mean, or even how the results have occurred. The manuscripts answers the question “what are the differences?” but not the questions “why are there differences?” or “how are there differences”. The majority of results are qualitative where quantitative comparisons are possible, and the quantitative data is presented far too late. This leaves the manuscript a little disappointing in its delivery. As a result, I recommend major revisions to this manuscript and resubmission.

Scope: The scope of this paper is too wide and not deep enough. From the title to the conclusions, the manuscript is mostly descriptive, with little detailed explanation. The manuscript doesn’t explore how the findings of the research project exist within the context of existing knowledge and literature, and therefore the contribution of this work to the wider research area is lost. Descriptions of non-tropical climate regimes is distracting and doesn’t add much to the paper compared to the explorations of tropical climate regimes. This could/should be a tropics focused paper. By orientating the paper to ask the question “how well do reanalysis datasets capture rainfall variability in the tropics?” this paper would have a much better and more precisely defined scope. This does not necessarily require additional data. The European Region could be retained as a control experiment with a dense, continuous weather station network. The introduction would need some substantial modification to reframe the research question. And some additional data analysis would be needed to understand differences in more detail and/or the causes behind differences But overall, the manuscript would be better for a reduced scope, and more detailed assessment.

Quantitative Assessments: There is also a lack of quantitative assessment. A lot of the work is very qualitative, and the data is not always presented well enough to assess whether the statements made by the authors are indeed true. The quantitative data is presented right at the end in section 4.5. This section, and table 3 need to be presented as one of the first results, and the numbers from it need including far more frequently in the text.

Additional Remarks:

  • Throughout: The phrase “Pacific ITCZ” should be used describe the ITCZ region used in the analyses. This removes confusion when talking about the actual ITCZ and ensures consistency between the text and figure 1.
  • Line 42: the density of measurements does not degrade back in time. There were far more weather stations 40-50 years ago than today.
  • Line 69: citation needed
  • Table 1: is the frequency column needed if all the datasets are monthly?
  • Figures in general: I assume the black boxes on the figures are a result of the particular file and typesetting and will be removed in the actual manuscript?
  • Line 463-465: This doesn’t seem right.
  • Line 466: less than 0.2mmday-1
  • Line 485-490: I think this needs more exploring. There must be more substantial literature either on the overestimation of above ocean rain by reanalaysis or the underestimation by gridded observation products.
  • Figure 4: Slight misalignment of the middle right panel
  • Figure 4 & 5: why are there eight categories between 0 and 10mm? This doesn’t make logical sense. Why not five or ten?
  • Figures 4 & 5: It’s hard to assess the differences between the control (observations) and test (reanalysis) datasets. The control datasets in green need to be more obvious. One idea would be to increase the saturation of those two products relative to the reanalysis datasets. Additionally/alternatively, it may be better to change the order of the datasets from alphabetical order to a category order (observations on the left, reanalysis on the right).
  • Line 507: You should be able to test this quantitatively. Is the low frequency anomaly enough to account for the offset?
  • Line 597-601: Again, this can be quantified.
  • Line 609: RMSD?
  • Figure 6: These panels need to have the same scale to properly assess the section from lines 539 to 623.
  • Section 539-623 is an example of where appropriate quantification is needed far more frequently. E.g., Line 536 is very vague.
  • Line 671: This is an oversimplification. The tropics can have huge annual cycles. Just that the effects are spatially averaged to produce reasonably consistent tropic wide rainfall.
  • Figure 10: These panels need to have the same scale to assess these changes.
  • Paragraph 722-731: An example of where significant quantification is needed.
  • Line 741: It’s not possible to really assess this claim with the data provided. I think Figure 11 needs an extra set of panels showing the seasonal cycle normalised to GPCP. While the absolute bias of JRA-55 and MERRA-2 might be higher in boreal summer, the proportional bias might not be (i.e it might always be a 1.2x overestimation).
  • End of section 4.4: Can these results be related to the observed differences between products described earlier in the text.
  • Section 4.5: This needs to be much earlier.
  • Line 780-783: This non-tropical work is distracting.

Round 2

Reviewer 1 Report

The authors have folowed motf of my comments and the paper has significantly improved. However, they have not provided a "clean" version and there are some slight mistakes that should be resolved, such as:

Line 354:  of the globe (see )---> see what???

Line 431: Table 2Table 2---> delete one of them

Table 2: delete (1) or specify it means dimensionless

Reviewer 2 Report

A thorough and considerate set of responses, and some significant improvements to the manuscript.

I only have a few typos/suggestions:

Line 343 (apart from)?

Line 354: (see )?

Line 431: repeat of Table 2

Line 441: delete the first and

Line 441: three

Figure 6-11: Panels could be labelled with text as well as a) b) c)
